# Central and peripheral clocks are coupled by a neuropeptide pathway in *Drosophila*

Mareike Selcho[1,*], Carola Millán[2,*,†], Angelina Palacios-Muñoz[2,*,†], Franziska Ruf[1], Lilian Ubillo[2], Jiangtian Chen[1], Gregor Bergmann[1], Chihiro Ito[1,†], Valeria Silva[2], Christian Wegener[1] & John Ewer[2]

Animal circadian clocks consist of central and peripheral pacemakers, which are coordinated to produce daily rhythms in physiology and behaviour. Despite its importance for optimal performance and health, the mechanism of clock coordination is poorly understood. Here we dissect the pathway through which the circadian clock of *Drosophila* imposes daily rhythmicity to the pattern of adult emergence. Rhythmicity depends on the coupling between the brain clock and a peripheral clock in the prothoracic gland (PG), which produces the steroid hormone, ecdysone. Time information from the central clock is transmitted via the neuropeptide, sNPF, to non-clock neurons that produce the neuropeptide, PTTH. These secretory neurons then forward time information to the PG clock. We also show that the central clock exerts a dominant role on the peripheral clock. This use of two coupled clocks could serve as a paradigm to understand how daily steroid hormone rhythms are generated in animals.

[1] Neurobiology and Genetics, Theodor-Boveri-Institute, Biocenter, University of Würzburg, Am Hubland, 97074 Würzburg, Germany. [2] Centro Interdisciplinario de Neurociencia de Valparaíso, Facultad de Ciencias, Universidad de Valparaiso, Gran Bretaña 1111, Valparaíso 2360102, Chile. * These authors contributed equally to this work. † Present addresses: Facultad de Artes Liberales y Facultad de Ingeniería y Ciencias, Universidad Adolfo Ibañez, Viña del Mar 2580760, Chile (C.M.); Departamento de Psiquiatría, Escuela de Medicina, Centro Interdisciplinario de Neurociencia, Pontificia Universidad Católica de Chile, Diagonal Paraguay 362, Santiago 8330077, Chile (A.P.-M.); Graduate School of Science, Kyoto University, Sakyo, Kyoto 606-8502, Japan (C.I.). Correspondence and requests for materials should be addressed to C.W. (email: christian.wegener@biozentrum.uni-wuerzburg.de) or to J.E. (email: john.ewer@uv.cl).

Circadian clocks impose daily periodicities to many behaviours and physiological processes in a wide variety of organisms. In animals, many tissues are known to contain circadian pacemakers. Whereas much is currently known about the molecular mechanisms that produce rhythmicity within circadian pacemakers, less is known about how the activity of these pacemakers is coordinated within an animal, even though loss of synchronization between pacemakers can contribute to circadian rhythm and sleep disorders, as well as metabolic and cardiovascular diseases[1–3]. In mammals, coordination is effected by the central pacemaker of the suprachiasmatic nucleus (SCN)[4] through poorly understood pathways, but which include neural, endocrine[5,6] and even thermal signals[7]. In insects, by contrast, most peripheral pacemakers are autonomous, and their synchronization is accomplished through exposure to common entraining signals such as light, which can penetrate the translucid exoskeleton, or temperature[8]. However, more complex relationships between central and peripheral clocks have also been reported. In *Drosophila*, for example, the clock in the peripheral pheromone-producing oenocyte cells can be set by pigment dispersing factor (PDF), the principal neuropeptide produced by the central clock in the brain[9]. An even more complex relationship has recently been reported for the fat body, where the cycling of some genes depends on the fat body clock whereas that of other genes occurs in response to peptidergic central brain signals mediated by the neuropeptide F (NPF) neuropeptide[10,11].

Here we investigated the mechanism that couples the central brain clock to the peripheral clock of the prothoracic gland (PG), which underlies the circadian control of adult emergence. The PG is the endocrine gland that produces the steroid moulting hormone, ecdysone (E), and previous work in *Drosophila*[12,13] and other insects[14,15] has shown that the activity of the PG clock depends on the brain clock. Together these two clocks restrict the time of adult emergence to a species-specific window within the day[16–18]. A long-standing presumption has been that the brain clock communicates with the PG clock via PDF[16]. In the larval brain of *Drosophila*, the PDF-expressing clock neurons known as 'lateral neurons' (LNs) project to the dorsal region of the brain and terminate in close proximity to the dendritic arbours of neurons that produce the prothoracicotropic hormone (PTTH), which controls ecdysone production by the PG[19–21].

Here we show that a similar anatomical relationship between the small ventral LNs (sLNvs), a critical subclass of central pacemaker neurons, and the PTTH neurons also exists in the pharate adult brain of the fruit fly. Yet, this connection is not effected by PDF but by short NPF (sNPF), a second neuropeptide produced by sLNv neurons[22]. We also show that PTTH neurons transmit the central brain timing information to the PG via the neuropeptide PTTH, acting on its receptor encoded by the *torso* gene[23]. Finally, by examining the consequences on the rhythmicity of emergence of separately altering the speed of the central versus the peripheral PG clock we show that the brain clock exerts a dominant influence on the PG clock. Thus, our findings identify the peptidergic pathway that connects the brain clock to the peripheral PG clock to regulate the timing of this critical behaviour, and demonstrate the hierarchical relationship between these two clocks. Our results are reminiscent of the control of the circadian rhythm of glucocorticoid (GC) production in mammals, which depends on functional clocks in the SCN and in the adrenal gland. In this case, the mechanisms through which central and peripheral clocks are coordinated to time steroid hormone action are still incompletely understood. The control of emergence in *Drosophila* may provide a genetically tractable model to uncover general mechanisms behind such daily endocrine rhythms.

## Results

**sLNv clock neurons are connected to PTTH neurons via sNPF.** The central clock of *Drosophila* includes around 150 clock neurons[24]. Of these, 8 pairs of ventral LNs (LNvs) express the neuropeptide PDF, which is key for the expression of behavioural circadian rhythmicity[25,26], including rhythmic adult emergence[16,17], and is one of the intercellular signals that coordinates the phase of the various clock neurons[27,28]. In addition, the prominent projections from the small LNvs (sLNvs)

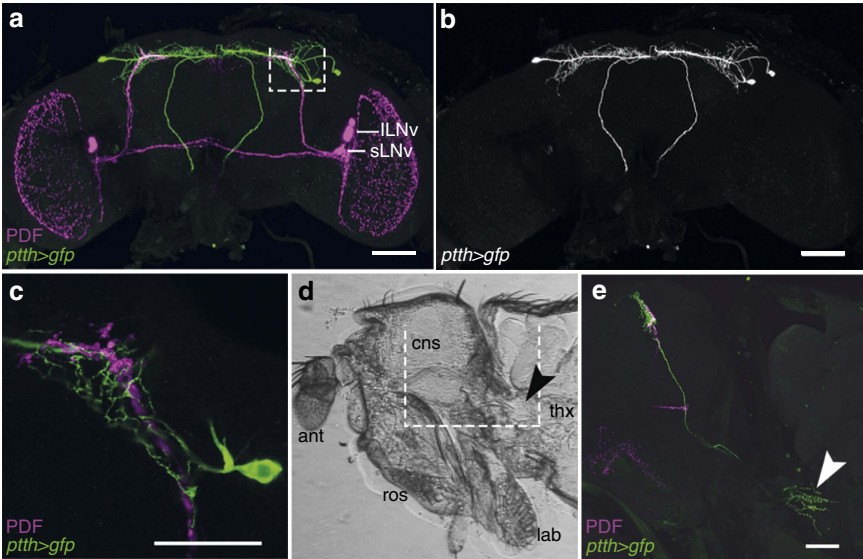

**Figure 1 | Anatomical relationship between PDF- and PTTH-positive neurons. (a)** PDF and *ptth*-gal4 expression in the fly brain before emergence. PDF-positive (magenta) and PTTH neurons (green) arborize in the superior protocerebrum. (**b**) Arborization pattern of PTTH neurons shown alone. (**c**) Higher magnification of a 1 μm Z-stack of the superior lateral protocerebrum boxed in **a**, showing the close proximity between the arborizations of PTTH neurons and the endings of the PDF-expressing sLNvs. (**d**) Sagittal view of the medial region of a pharate fly head and anterior thorax. (**e**) Confocal projection of the white box shown in **d**. PTTH neurons innervate the PG where they form a dense varicose mesh of putative release sites (arrowhead). ant, antenna; lab, labellum; cns, central nervous system; ros, rostrum; thx, thorax. Scale bars in **a**,**b** and **e**: 50 μm, in **c**: 25 μm.

to the superior protocerebrum occur in close proximity to neurons known to control specific behaviours, suggesting that PDF may also be a clock output signal. In the pharate adult brain in particular, it overlaps with the dendritic field of the PTTH producing neurons (Fig. 1a–c), which regulate the timing of every insect moult[29]. The PTTH neurons project to the PG (Fig. 1d,e), where local release of PTTH causes the synthesis of E (ref. 21). Since the levels of ecdysone must fall below threshold levels in order for emergence to occur[30,31], this anatomical arrangement suggests that PDF, acting on PTTH neurons, could provide a pathway through which the central clock could influence the PG clock and gate the time of adult emergence. Yet, despite these auspicious observations, we found that PTTH neurons did not respond to PDF application. The PDF receptor (PDFR) is a secretin receptor-like G protein-coupled receptor[32] and signals through cyclic adenosine monophosphate (cAMP; ref. 33). Challenging with PDF larval or pharate adult brains expressing the genetically encoded cAMP sensor, Epac1-camps[33], did not cause a decrease in FRET signal in PTTH neurons (Fig. 2a,c), which is expected[33,34] in response to an increase in cAMP, except when PDFR was ectopically expressed in these neurons (Fig. 2b,c). Consistent with this observation, downregulating PDFR in PTTH neurons did not affect the rhythmicity of emergence (Fig. 3b,f; compare with control, Fig. 3d,f). These results suggest that the aberrant timing of eclosion expressed by *pdf* null mutants[16] occurs because the lack of this neuropeptide causes the brain clock itself to become arrhythmic. Consistent with this, residual rhythmicity in *pdf* null mutants is apparent in the first 2 days of DD (Supplementary Fig. 1a, asterisks). This situation mirrors the one observed for locomotor activity[35] and suggests that PDF's role in the control of eclosion is limited to the coordination of the various pacemakers in the fly brain[27,28]. By contrast, we found that sNPF, a second neuropeptide produced by the PDF-positive sLNvs (ref. 22), caused a rapid decrease in the fluctuations in free calcium in PTTH neurons (Fig. 2d–f; Supplementary Movie 1), reminiscent of sNPF-induced suppression of spontaneous calcium waves in larval motoneurons[34]; no response was observed when we monitored changes in cAMP levels in PTTH neurons (Supplementary Fig. 2). Importantly, optogenetic activation of PDF neurons using the orthogonal LexA/LexAop expression system mimicked the reduction in cytosolic calcium levels induced by sNPF in the PTTH neurons (Fig. 2g–i; Supplementary Movie 2), suggesting that sLNv neurons could inhibit PTTH neurons via sNPF; no such response was obtained in flies lacking the *pdf*-LexA driver (*ptth*-Gal4 > GCaMP6m, > LexAop-ChR-XXL; Supplementary Movie 3). Furthermore, RNAi-mediated knockdown of the sNPF receptor specifically in PTTH neurons abolished the circadian rhythmicity of emergence (Fig. 3c,f; compare with control, Fig. 3d,f). These results indicate that the sNPF peptide is the principal circadian signal to PTTH neurons, and that this signal could be provided directly by sLNv neurons.

**PTTH transmits timing information to the PG**. We next investigated how the PTTH neurons transmit the timing information from the brain clock to the PG. PTTH neurons are known to produce the neuropeptide, PTTH, but other co-localized transmitters or peptides could be involved. Nevertheless, we found that knockdown of the PTTH neuropeptide in PTTH neurons also rendered emergence arrhythmic (Fig. 3e,f; compare with control, Fig. 3d,f); the same phenotype was obtained following targeted killing (Supplementary Fig. 3a,e) or electrical silencing of PTTH neurons (Supplementary Fig. 3b–e). These results demonstrate that the PTTH neurons provide, via the PTTH neuropeptide, the brain signal that gates the time of adult

emergence following light entrainment. This signal is likely required for circadian rhythmicity itself versus simply coupling the activity of the brain clock to the output pathway, since cultures entrained using a warm/cold (WC) temperature regime then also showed an arrhythmic pattern of emergence under conditions of constant darkness and temperature (DD + CC regime; Fig. 4a,j; compare with controls, Fig. 4b,c,j). Interestingly, such cultures eclosed rhythmically under DD + WC (Fig. 4d,j) and LD + CC (Fig. 4g,j) conditions (as did the corresponding controls, Fig. 4e,f,j and Fig. 4h,i,j, respectively). This could occur because the clock in the PG (see below) is directly entrained by light and temperature cycles; indeed, the PG of other insects has been shown to contain a light-entrainable and temperature compensated clock[36,37]. Although PTTH is required for circadian rhythmicity of emergence, the PTTH timing signal seems to be independent of rhythmic gene expression, as we were unable to detect circadian or ultradian fluctuations of *Ptth* messenger RNA levels in pharate heads (Supplementary Fig. 5). This suggests that the timing information from the central clock may be encoded in the PTTH neurons by a different mechanism, such as rhythmic peptide release.

To investigate whether PTTH transmits circadian timing information directly to the PG, we interfered with PTTH transduction in the PG. The receptor for PTTH is the tyrosine kinase-coupled receptor encoded by *torso*, which is expressed in the PG[23], as are the elements downstream of *torso* (Supplementary Fig. 6a). We found that knockdown of *torso* (Supplementary Fig. 6c)[23] and of its intracellular signalling pathway specifically in the PG eliminated the circadian rhythmicity of emergence (Fig. 3g–j, l; compare with control, Fig. 3k,l). Interestingly, no residual rhythmicity was observed when *torso* was knocked down in the PG (Supplementary Fig. 1b), suggesting that the PG clock is stopped when PTTH signalling to the PG is disrupted. Consistent with this, knockdown of *torso* in the PG also eliminated the molecular oscillations in the PG monitored using the *cre*-luciferase reporter[38] (Fig. 3m) as well as a *period*-luciferase reporter (Supplementary Fig. 7). Similarly, knockdown of *torso* using the *timeless*-gal4 driver, which is expressed in brain pacemaker neurons as well as in the PG (ref. 39; Fig. 5a–c), also rendered arrhythmic the daily pattern of emergence (Fig. 6a,c). Importantly, normal rhythmicity was restored in these animals when *torso* RNAi expression was excluded from the PG (Fig. 6b,c; cf. Fig. 5d–f), indicating that the PG is the only tissue to which PTTH conveys the circadian signal from the central clock, even though *torso* is also expressed in other tissues at this time (Supplementary Fig. 6b). Finally, knocking down *torso* in the PG (Supplementary Fig. 8a,e) or in other clock tissues (Supplementary Fig. 8b–e) did not affect the circadian rhythm of locomotor activity in adult flies.

**The brain clock exerts dominant influence over the PG clock**. Our findings show that the central clock transmits circadian timing information to the peripheral PG clock via PTTH. To investigate the relative importance of the central versus the PG clock in the control of the circadian rhythmicity of adult emergence, we investigated the consequences of genetically stopping these clocks on the circadian gating of eclosion. As shown in Fig. 7, we found that stopping the brain clock (Fig. 7b,e) or the PG clock (Fig. 7c,e) by overexpressing the dominant form of the *cycle* gene (*cyc[Δ901]* (ref. 40)) renders arrhythmic the pattern of adult emergence (as did stopping all clocks, Fig. 7a,e). A similar result was obtained by overexpressing the *timeless* gene in all clock tissues (including the brain) except for the PG (Supplementary Fig. 9b,i) or only the PG clock (Supplementary

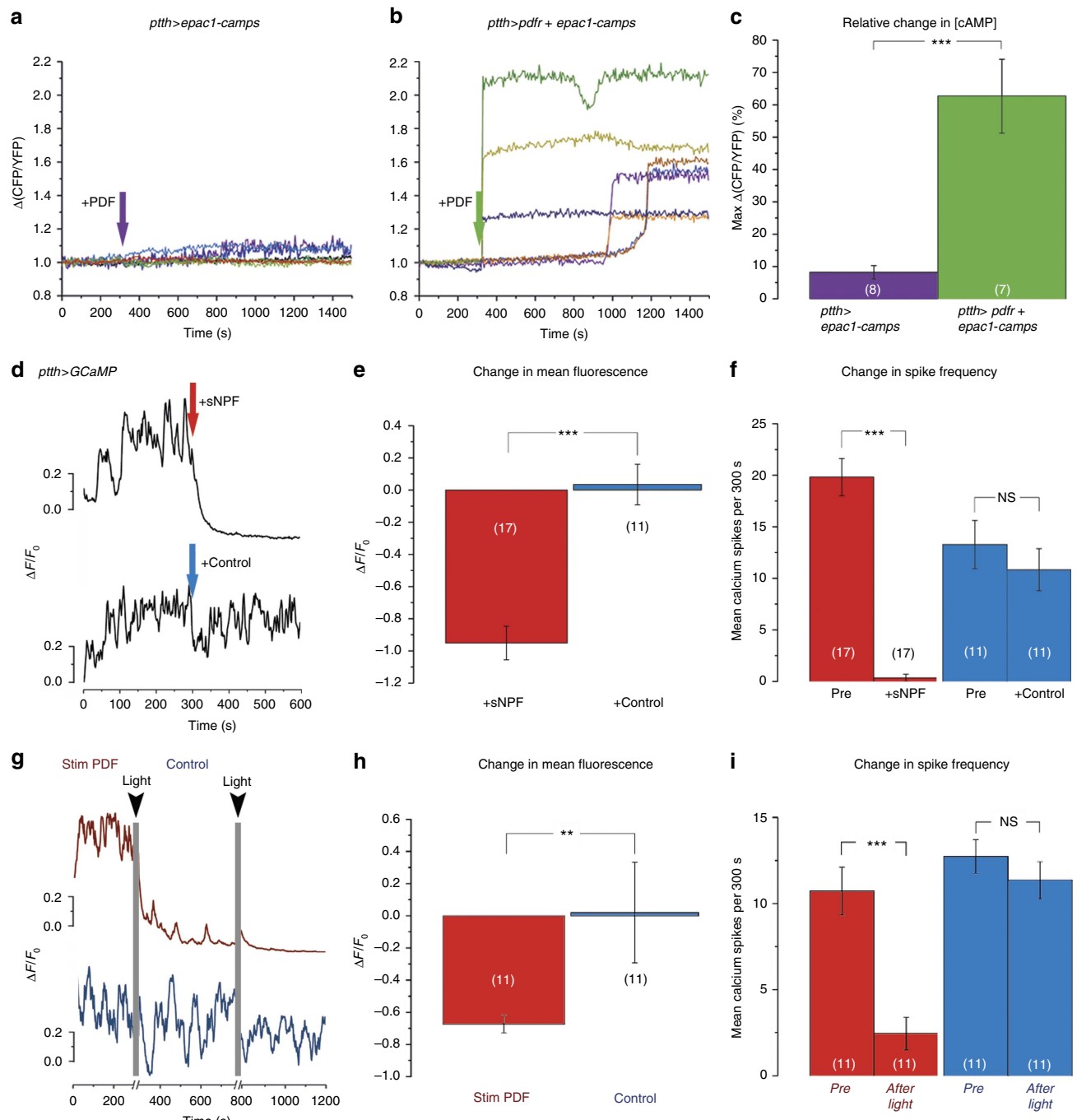

**Figure 2 | PTTH neurons respond to sNPF but not to PDF. (a,b)** Bath application of PDF ($10^{-5}$ M) did not elicit an increase in cAMP levels in PTTH neurons (**a**) unless they ectopically expressed PDFR (**b**); each trace represents the response from a single ROI of PTTH cell bodies ($N = 8$ ROIs from 7 pharate adult brains for **a**, 7 ROIs from 5 pharate adult brains for **b**). (**c**) Summary of results shown in **a,b**, indicating the average maximum response upon PDF application ($\pm$ s.e.m.). (**d**) Bath application of sNPF ($10^{-5}$ M; red arrow) caused rapid reduction in spontaneous calcium oscillations in PTTH neurons (upper trace); no such response occurred when challenged with solvent alone (control; blue arrow, lower trace). (**e,f**) Average ($\pm$ s.e.m.) fluorescence change (**e**) and change in spike frequency (**f**) of GCaMP6 fluorescence induced by sNPF (red) versus control (blue) ($+$sNPF $N = 17$ ROIs from 11 pharate adult brains; control: $N = 11$ ROIs from 8 pharate adult brains). See also Supplementary Fig. 2. (**g**) Light stimulation of brains in which PDF neurons express ChR2-XXL caused a reduction in spontaneous calcium oscillations in PTTH neurons (red trace; time of light stimulation indicated by grey vertical lines); no such effect was obtained when the *pdf*-LexA transgene was omitted (blue trace; control). (**h,i**) Average ($\pm$ s.e.m.) change in intensity (**h**) and change in spike frequency (**i**) of GCaMP6 fluorescence induced in PTTH neurons by light stimulation in flies expressing ChR2-XXL in PDF neurons (red) versus control (blue). (Stim PDF: $N = 11$ ROIs from 6 pharate adult brains; control: $N = 11$ ROIs from 7 pharate adult brains). (NS $= P > 0.05$; **$P < 0.01$; ***$P < 0.001$; *t*-test: **e;f;i** blue; Wilcoxon rank sum test: **c;h;f, i** red).

Fig. 9d,i); by stopping the clock in PDF-expressing neurons using *cyc[Δ901]* (Supplementary Fig. 9c,i); and in the *pdf* null allele (Supplementary Fig. 1a). These results are consistent with

previous reports[16–18], but differ from intriguing findings recently reported by Di Cara and King-Jones[41], which show that specifically stopping the PG clock is mostly lethal due to a

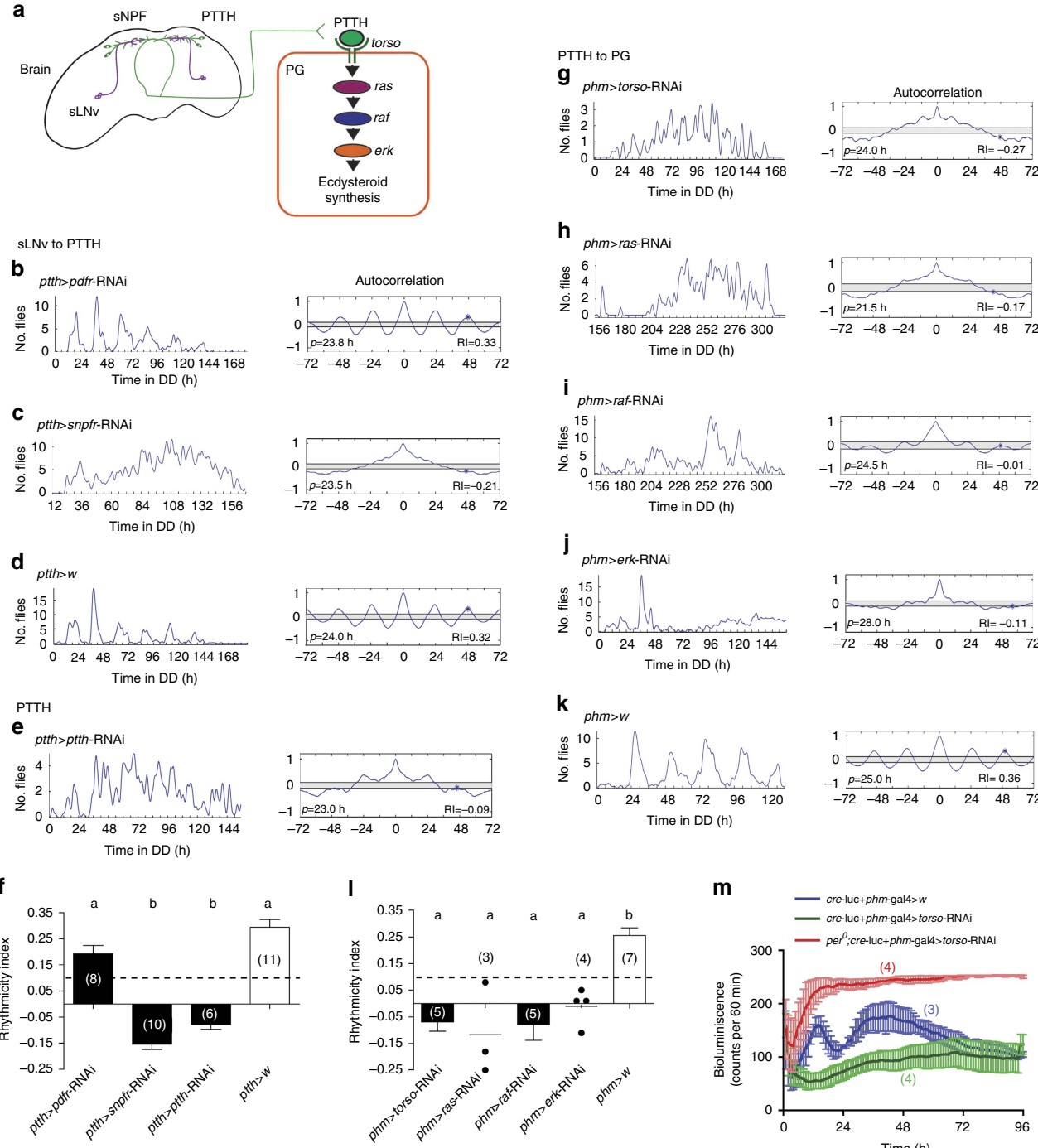

**Figure 3 | Brain and PG clocks are coupled by sNPF and PTTH.** (**a**) Diagram showing the neuronal and molecular components that couple the brain clock (left) to the PG clock (right). sLNv neurons (purple line) contain sNPF and project dorsally to the vicinity of PTTH neurons (green). PTTH acts on the PG to activate the TORSO intracellular transduction pathway. (**b**) Knockdown of PDFR in PTTH neurons did not alter the circadian rhythmicity of emergence; by contrast, knockdown of sNPF receptor in these neurons rendered arrhythmic the pattern of adult emergence (**c**). Records show time course of emergence of a single population in DD (left) and corresponding autocorrelation analysis (right); principal periodicity and associated RI is indicated. (**e**) Reduction of PTTH neuropeptide in PTTH neurons eliminated the circadian rhythmicity of emergence. (**f**) Average RI values ( ± s.e.m.) for results shown in **b**,**c**,**e** and in control **d**; dashed lines mark RI cutoff value of 0.1. Different letters indicate statistically different groups ($P < 0.05$; one-way ANOVA, Tukey's *post hoc* multiple comparison analyses). Numbers in parenthesis indicate number of separate experiments. (**g**–**j**) Knockdown of PTTH transduction pathway in the PG caused the expression of an arrhythmic pattern of adult emergence. (**l**) Average RI values ( ± s.e.m.) for results shown in **g**–**j** and in control **k**, shown as described for **f**; individual values are indicated when $N < 5$ and average indicated by short horizontal line. Flies bearing only UAS-RNAi transgenes express normal rhythmicities of emergence (Supplementary Fig. 4). See also Supplementary Fig. 3. (**m**) Knockdown of *torso* expression in the PG eliminated circadian fluctuations of *cre*-driven bioluminescence in the PG. Values plotted correspond to average ± s.e.m.; numbers in parenthesis indicate number of records averaged. See also Supplementary Fig. 7. In all experiments RNAi knockdown was enhanced by co-expression of *dcr2*.

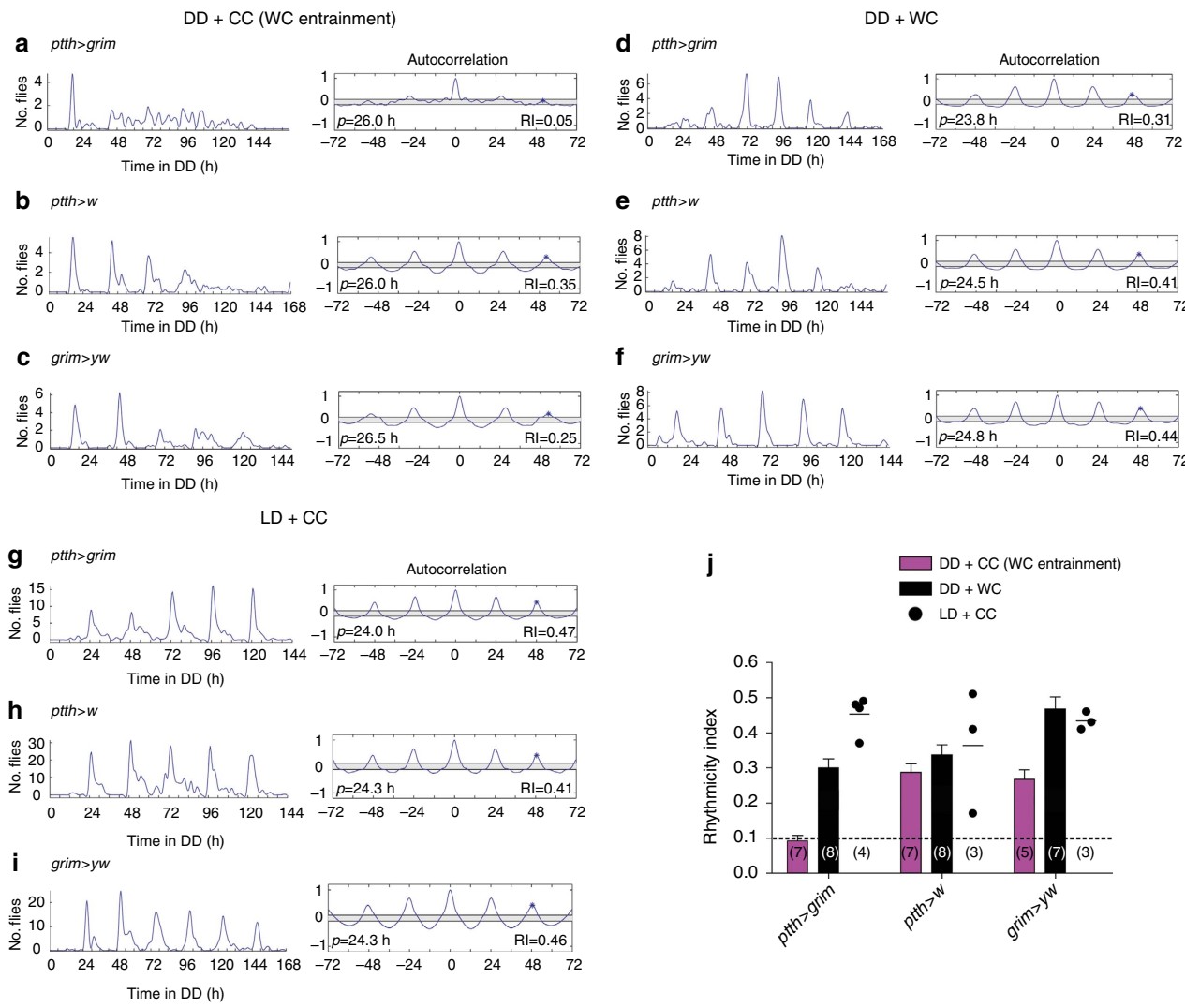

**Figure 4 | PTTH neurons are required for entrainment by temperature cycles.** (**a–c**) Left: pattern of emergence in DD from cultures entrained in DD with a 12 h:12 h 25 °C/16 °C (WC) temperature cycle in which PTTH neurons were selectively killed (**a**) and in controls (**b,c**); Right: corresponding autocorrelation analysis with value of RI indicated. (**d–f**) Left: pattern of emergence under DD + WC regime when PTTH neurons were selectively killed and in controls (**e,f**). (**g–i**) Left: pattern of emergence under LD + CC regime when PTTH neurons were selectively killed (**g**) and in controls (**h,i**). (**j**) Average RI ( ± s.e.m.) for genotypes and conditions shown in **a–i**; dashed line marks RI cutoff value of 0.1. Numbers in parenthesis indicate number of separate experiments. Individual values are indicated when N < 5 and average indicated by short horizontal line.

disruption in ecdysone synthesis; how these results can be reconciled with our and previous findings awaits further investigation.

Although our results (Fig. 7) and those of others[16–18] show that functional clocks are required in both the brain and the PG for a circadian pattern of emergence, they do not inform on the hierarchical relationship between these two clocks because we do not understand the mechanism by which the brain clock influences the PG clock. In particular, we do not know the consequences on the PG clock of stopping the brain clock. The PG is known to house an autonomous, light-entrainable and temperature compensated clock[12,13,31,37], yet stopping the brain clock could result in the production of a dominant PTTH signal that stops the PG clock, obscuring any autonomous action that the PG clock could exert in the absence of a central clock. For this reason, rather than stopping the central or the peripheral PG clock, we explored their hierarchical relationship by investigating the consequences on eclosion rhythmicity of separately manipulating their speed using targeted expression of *doubletime* kinase alleles[28]. Using this approach, we found that

slowing down the brain clock caused a lengthening in the periodicity of emergence that was indistinguishable from that obtained when the speed of all clocks was similarly manipulated (Fig. 8b versus Fig. 8a; Fig. 8d). A comparable result was obtained when the clock in the brain was selectively sped up (Fig. 8f versus Fig. 8e; Fig. 8h), although a slight contribution from the PG could be detected in this case. Nevertheless, such a contribution is comparatively minor, since slowing down (Fig. 8c,d) or speeding up (Fig. 8g,h) only the PG clock using the strong PG gal4 driver, *phm*-gal4 (Fig. 5g–i), did not cause statistically significant changes to the periodicity of emergence. Thus, the timing of emergence is controlled by two coupled clocks, in which the brain clock plays a dominant role as compared to the PG clock, fully supporting the classical 'coupled-oscillator model' of brain and PG clock as 'master' and 'slave' clocks, respectively[42].

## Discussion

The emergence of the adult fly is controlled by the circadian clock, which restricts the time of eclosion to a species-specific

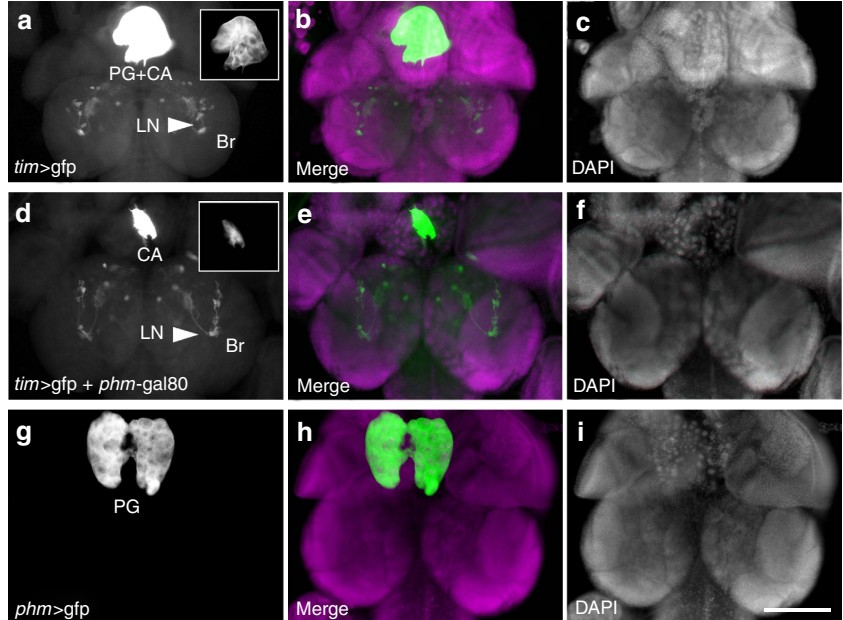

**Figure 5 | *phm*-gal80 effectively suppresses *tim*-gal4 expression in the PG.** Pattern of GFP (left; **a,d,g**) and corresponding DAPI (right; **c,f,i**) and merged images (center; **b,e,h**) of pre-pupal brains (Br) expressing GFP under the control of *tim*-gal4 (**a**), *tim*-gal4 + *phm*-gal80 (**d**) and *phm*-gal4 (**g**). Gain in **a,b** was set to visualize expression within the brain; arrowheads indicate LNs. Insets in **a,b** show expression in PG using the same gain used for **g**. In the ring gland, *tim*-gal4 drives expression in the PG and the *corpora allata* (labelled CA in **a,d**). Expression in the PG is suppressed when the *tim*-gal4 driver is combined with *phm*-gal80 (**d**). Scale bar (for **a**–**i**): 100 μm.

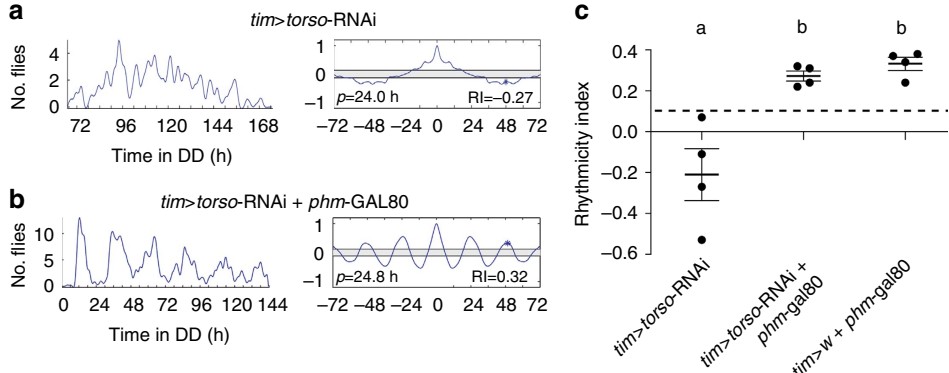

**Figure 6 | *torso* is required only in the PG for circadian rhythmicity of eclosion.** (**a**) Driving *torso* RNAi in all clocks using *tim*-gal4 driver rendered the pattern of emergence arrhythmic. Left: pattern of eclosion in DD; right: autocorrelation analysis of record, with dominant periodicity and value of RI indicated. (**b**) Suppressing *tim*-gal4 action in the PG combining *tim*-gal4 with *phm*-gal80 restored the circadian rhythm of eclosion indicating that *torso* function is only required in the PG clock for a circadian rhythmicity of emergence. (**c**) Individual values and average RI ( ± s.e.m.) for genotypes shown in **a,b**; dashed line marks RI cutoff value of 0.1. Different letters indicate statistically different groups (*P* < 0.05; one-way ANOVA, Tukey's *post hoc* multiple comparison analyses). Numbers in parenthesis indicate number of separate experiments. In all experiments RNAi knockdown was enhanced by co-expression of *dcr2*.

time window. This control depends on the presence of a functional clock in the brain and in the PG. Here we have identified the pathway through which the central brain clock exerts its influence on the peripheral PG clock (cf., Fig. 3a). The PDF neuropeptide plays a key role in coordinating the activity of the network of pacemaker neurons of the central clock[26,43] and also plays a role in setting the phase of some peripheral clocks[9]. Yet, although it has long been assumed that the brain would influence the PG clock via PDF, we show here that the PTTH neurons receive central clock input via the inhibitory sNPF neuropeptide; PTTH neurons then transmit this time information to the PG via the PTTH neuropeptide. This arrangement joins other cases where LN clock neurons communicate with the periphery using neuropeptides other

than PDF, as occurs for some cycling genes in the fly's fat body[10], which is mediated through NPF. In line with sNPF's inhibitory action on calcium activity of the PTTH neurons in pharate adults, ablation of PDF neurons allows a stronger and faster response to light stimulation in larval PTTH neurons[19]. It will be interesting to establish whether the use of sNPF-mediated inhibition as a clock-to-PTTH neuron signal is a general feature throughout the insects, as close proximity of clock and PTTH neuronal termini has been found in many hemi- and holometabolous insects[44–46].

By selectively speeding up the brain or the PG clock we show that the brain clock exerts a dominant action on the PG clock. Assuming that sNPF is temporally co-released with PDF, our findings imply that the central clock's contribution to the gating

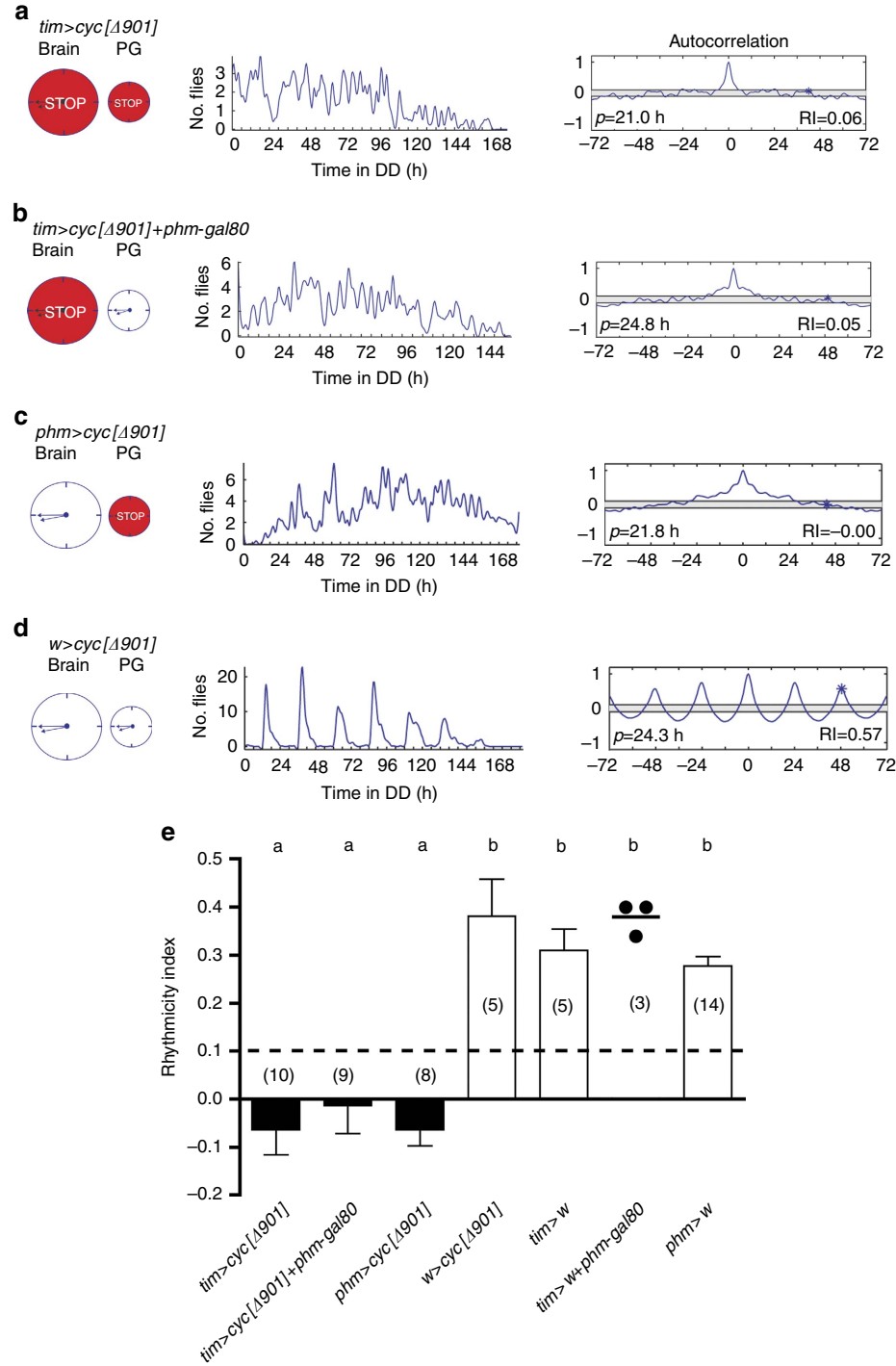

**Figure 7 | Stopping clock in brain and/or PG leads to arrhythmic adult emergence.** Pattern of emergence when all clocks (**a**), the brain clock (**b**) or the PG (**c**) clock, have been stopped. Schematic shown on the left represents the brain and PG clocks (large and small circle, respectively); red indicates which clock has been stopped; middle: pattern of eclosion in DD; right: autocorrelation analysis of record, with dominant periodicity and value of RI indicated. (**e**) Average RI ( ± s.e.m.) for genotypes shown in **a**–**d** and for controls; individual values are indicated when $N < 5$ and average indicated by short horizontal line. Dashed line marks RI cutoff value of 0.1. Different letters indicate statistically different groups ($P < 0.05$; one-way ANOVA, Tukey's *post hoc* multiple comparison analyses). Numbers in parenthesis indicate number of separate experiments. *phm*-gal80 effectively suppresses *tim*-gal4 expression in the PG (cf. Fig. 5d). (Strictly speaking, in (**b**) gene expression is driven in all clocks except the PG. However, since circadian rhythmicity of emergence depends only on the clocks in the brain and in the PG, such experiment is equivalent to driving gene expression only in the brain.)

of eclosion would be mediated through an inhibition of steroidogenesis starting in the early morning, when PDF release is believed to be maximal[47]. This signal would then cause the ecdysone titer to drop below the required threshold[30,31], resulting in the commitment to emerge during the next gate to be made

around lights-off[48] and causing the gate to open starting at dawn of the following day.

In the bug, *Rhodnius*, moulting is accompanied by dramatic circadian oscillations in ecdysone titers[31,49]. Interestingly, whereas these titers in general rise during the dark phase

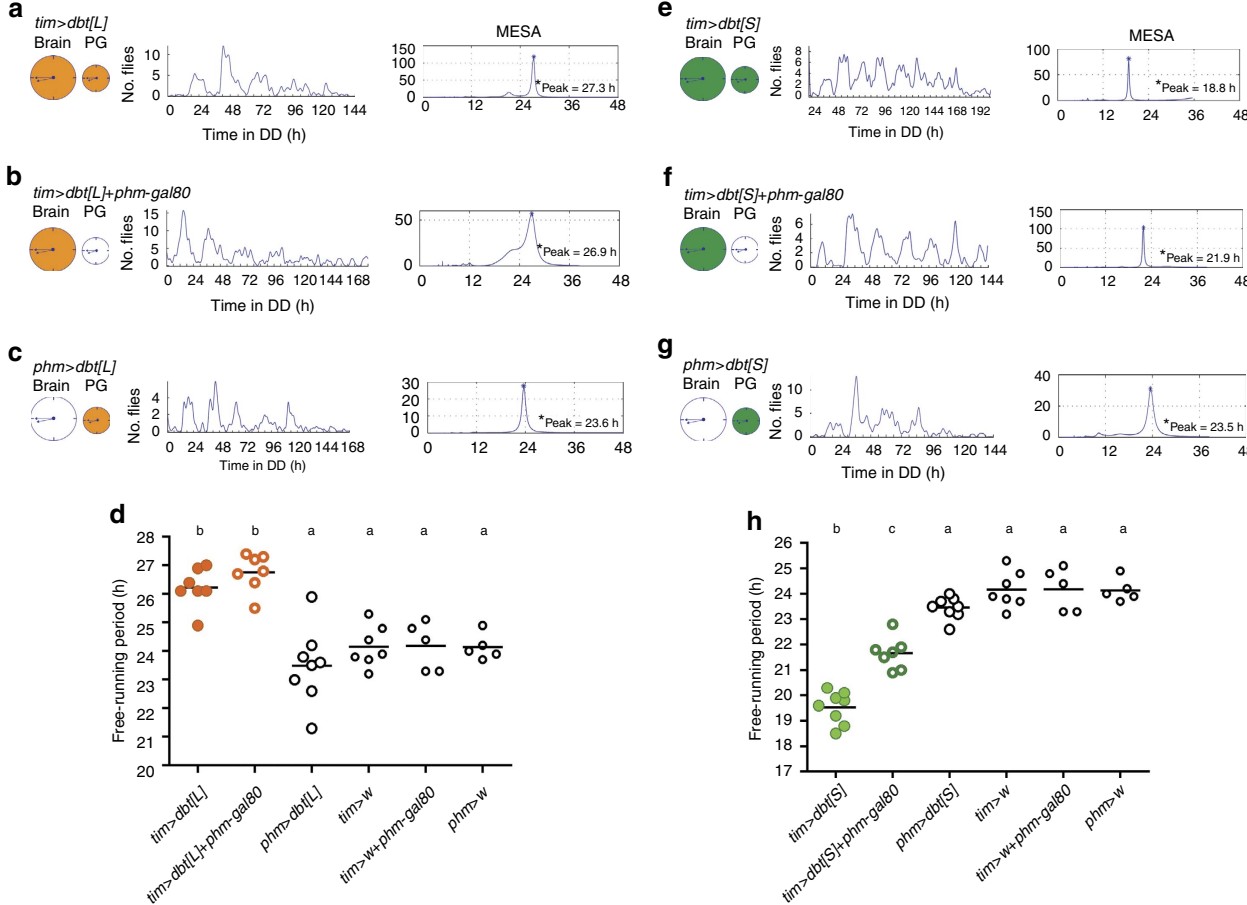

**Figure 8 | Hierarchical relationship between the brain clock and the PG clock.** (**a–c**) Pattern of emergence when all clocks (**a**) or only the brain (**b**) or the PG (**c**) clock have been slowed down. Schematic shown on the left represents the brain and PG clocks (large and small circle, respectively); orange colouring indicates which clock has been slowed down; middle: pattern of eclosion in DD; right: MESA analysis of record, with dominant periodicity indicated. (**d**) Average periodicities ($\pm$ s.e.m.) for genotypes shown in **a–c** and for controls; circles indicate periodicity for each separate experiment ($N = 5$–8); average is indicated by horizontal line; different letters indicate statistically different groups ($P < 0.05$; one-way ANOVA, Tukey's *post hoc* multiple comparison analyses). (**e–g**) Pattern of emergence when all clocks (**e**), or only the brain (**f**) or the PG (**g**) clock have been sped up (indicated in green). (**h**) Average periodicities ($\pm$ s.e.m.) for genotypes shown in **e–g** and for controls, represented as described in **d**. *phm*-gal80 effectively suppresses *tim*-gal4 expression in the PG (cf. Fig. 5d). (Strictly speaking, in (**b,f**) gene expression is driven in all clocks except the PG. However, since circadian rhythmicity of emergence depends only on the clocks in the brain and in the PG, such experiment is equivalent to driving gene expression only in the brain).

of the light:dark cycle, they drop during the final night before emergence; this final drop would then lead to the opening of the emergence gate. Furthermore, injections of 20E destroy the gating of emergence[31], consistent with the notion that gating is regulated by ecdysone. Nevertheless, this picture differs from that obtained in other insects. In particular, ecdysone titers show at most shallow circadian oscillations in the moth, *Manduca sexta*[30], and injections of 20E do not destroy the gating of adult emergence but instead cause animals to emerge later during the same gate or to eclose during a later gate[50]; likewise, no circadian oscillations in 20E titers have been described during *Drosophila* metamorphosis[51,52]. Thus, the specific mechanism may differ among insect species. In particular, the entirety of metamorphosis of *Drosophila* lasts only about 100 h, which may be too short a duration for timing of emergence to be controlled through a circadian regulation of ecdysone titers. Nevertheless, it could be that in all insects the time when the circadian clock exerts its critical influence is during the final day of the moult. Indeed, regardless of whether ecdysteroids express circadian oscillations in titers, the commitment to emerge occurs during the last day before emergence (*Rhodnius*[53], *Manduca*[30] and *Drosophila*[48]).

Although we have identified the neuropeptides that connect the brain clock to the PG clock, many questions remain. In particular, we do not understand the relationship between PTTH signalling and 20E production. Indeed, although PTTH synthesis is under clock control (at least in the adult[54]), its messenger RNA does not cycle with circadian periodicity in the pharate adult (Supplementary Fig. 5), the larva[21] or the adult[54]. It will also be important to determine how PTTH signalling affects the PG clock. It is especially intriguing that both brain and PG clocks are necessary for a circadian rhythmicity of emergence (Fig. 7, Supplementary Figs 1 and 9), yet interfering with PTTH signalling in the PG eliminates PG clock cycling (Fig. 3m; Supplementary Fig. 7). This result is reminiscent of the situation described for the clock control of GC production by the mammalian adrenal gland. In this case circadian cycling in GC titers also depends on a functional central clock (housed in the SCN) as well as a functional peripheral clock in the adrenal gland[3], which gate the sensitivity to corticotropin-releasing hormone. The underlying mechanisms are incompletely understood but involve the regulation of cholesterol transport via clock control of Steroidogenic Acute Regulatory protein (StAR) expression. Interestingly, the GC rhythms affect the

cycling of clock genes in other peripheral clocks, revealing that the adrenal gland clock may play a more dominant role among peripheral clocks. By analogy, and especially considering that 20E is also a steroid hormone with potent genomic effects[55], it is possible that the PG clock could exert a similarly dominant influence on other peripheral clocks.

Ecdysone feeds into the molecular clockwork by modulating *clockwork orange* and *clock/period* expression via the ecdysone receptor and the downstream targets, *Eip75* (ecdysone-induced protein 75) and microRNA *let-7* (refs 54,56,57). Interestingly, *Eip75B* is the fly homologue of REV-ERB α and REV-ERB β, which are key regulatory elements of the core molecular clock of mammals[6,29]. This notable conservation between mammals and *Drosophila* in the functioning and steroid feedback of the circadian clock extends the previously noted conservation at the level of the pacemaker itself[58] and at the level of the intra-pacemaker coordination by peptides activating the same Family B1 GPCR signalling pathway vasoactive intestinal polypeptide (VIP), and PDF, in mice and flies, respectively[26,59]). This could mean that the genetically tractable circadian system of the fly could also be used to understand how the different mammalian clocks are coordinated. It will also be interesting to determine whether similar deep homologies extend to the actions of the sNPF signalling pathway and those of its vertebrate homologue, prolactin-releasing peptide[60], which has been implicated in the control of mammalian circadian rhythmicity and/or sleep regulation[61].

## Methods

**Fly rearing and fly stocks.** All flies were reared at room temperature (21–25 °C) on standard cornmeal/molasses/yeast food medium. All stocks with the exception of *phm-gal80* and *LexAop-Channelrhodopsin2-XXL* (see below) have previously been described and are listed in Supplementary Methods. All genotypes involving the use of UAS-RNAi transgenes included a copy of a UAS-*dcr2* transgene. When performing RNAi knockdown, several UAS-RNAi lines were tested and the one producing the most extreme phenotype was used (see Supplementary Methods).

**phm-gal80.** A 1.2 Kb DNA fragment containing the *phm* promotor was PCR-amplified using primers (see Supplementary Table 1) that included *Bgl*II (5′) and *Spe*I (3′) restriction sites, and cloned into the pGEM-T easy vector (Promega, WI, USA). It was then sequenced for verification, cut out using *Bgl*II and *Spe*I, and cloned into pPelican *P*-element vector[62] cut with *Bam*HI and *Spe*I (first purified away from the fragment containing the HSP70 TATA and eGFP DNA); this last step was completed by Genewiz (NJ, USA). DNA from sequence-verified clone was then sent for transformation to BestGene (Chino Hills, CA, USA).

**LexAop-Channelrhodopsin2-XXL.** The *Channelrhodopsin2-XXL* (ChR2-XXL) gene[63] (kind gift of Robert Kittel, Tobias Langenhan, and Georg Nagel, University of Würzburg), was PCR-amplified using Gateway primers (see Supplementary Table 1) and a Phusion High-Fidelity polymerase (Thermo Scientific), then sequenced for verification. The amplified sequence was introduced into the Gateway destination vector pLOT-W (ref. 64). The resulting LexAop-ChR2-XXL construct was introduced into the germline of *w*[1118] flies using standard *P*-element transformation by BestGene (Chino Hills, CA, USA). Several independent transformant lines were obtained. The functionality of the construct was tested by panneuronal expression using *elav*-LexA and subsequent activation by blue light, which induced a paralysis not seen in controls.

**RT-PCR.** Larval CNSs, whole prepupae and PGs were dissected under ice-cold phosphate-buffered saline (PBS). To aid PG visualization, larvae expressing GFP under the control of *phm*-gal4 were used. Total RNA was extracted using Trizol reagent (Invitrogen, Carlsbad, CA, USA) following manufacturer's instructions, and resuspended in 20 μl of RNAse-free water.

cDNA synthesis was carried out using SuperScript II Reverse Transcriptase (Invitrogen) following manufacturer's instructions. RNA of 1 μg was first treated for 15 min at room temperature with 1 μl DNAse I (2 U μl$^{-1}$), 10 × DNAse I Buffer (100 mM Tris, pH 7.5; 25 mM MgCl$_2$; 5 mM CaCl$_2$); DNAse I was then inactivated adding 1.5 μl 25 mM EDTA and incubating for 10 min at 65 °C. Oligo-dT (0.5 μg) and dNTP's (10 mM each) were then added to each sample, incubated at 65 °C for 5 min, then placed on ice. Four microlitres of 5 × First-Strand Buffer (250 mM Tris-HCl, pH 8.3; 375 mM KCl; 15 mM MgCl$_2$), 2 μl DTT (100 mM) and 1 μl RNAseOUT RNAse inhibitor (40 units μl$^{-1}$) were then added and samples

incubated for 2 min at 42 °C; finally, 1 μl of Reverse Transcriptase was added and the reaction incubated at 42 °C for 50 min, followed by 15 min at 70 °C. Control reactions were processed in parallel except that the Reverse Transcriptase was omitted.

To analyse gene expression, cDNA's were PCR-amplified using gene specific primers (see Supplementary Table 1); *rp49* was used as internal control. A 50 μl reaction mix included 5 μl 10 × buffer, 1.5 μl MgCl$_2$ (50 mM), 1 μl dNTP's (10 mM), primers (10 μM), 1 μl of cDNA and 0.2 μl Platinum High Fidelity Taq Polymerase (Invitrogen). The PCR programme consisted of 5 min at 95 °C, followed by 35 cycles of 30 s at 95 °C, 30 s at 55 °C, 60 s at 72 °C and finally 10 min at 72 °C.

Total RNA was extracted from pharate and 0–3 days old adult *Canton-S* flies using the Quick-RNA MicroPrep Kit (Zymo Research, Irvine, USA) according to manufacturer's instructions. Central brains, optic lobes and retinas (15 flies per group) were dissected in HL3.1 medium[65], and transferred to 300 μl RNA lysis buffer on ice. Additionally, the gonads of 10 adult males and females, as well as the complete abdomen of 10 male and female pharate flies were collected in 300 μl RNA lysis buffer on ice. As positive control, brains with ring glands from *Canton-S* L3 wandering larvae were collected and similarly treated. Tissues were homogenized with a plastic pestle. Total RNA was eluted in 8 μl RNAse-free water.

For cDNA synthesis, the QuantiTect Reverse Transcription Kit from Qiagen (Venlo, Netherlands) was used. All steps were performed following the manufacturer's protocol. Genomic DNA was removed by adding 1 μl of gDNA wipeout buffer to 6 μl of the eluted RNA. Following incubation at 42 °C for 2 min, the samples were placed for 2 min at 4 °C and 3 μl of a mastermix composed of 2 μl RT Buffer, 0.5 μl RT Primer Mix and 0.5 μl reverse transcriptase was added. Reverse transcription was performed for 30 min at 42 °C, followed by 3 min at 95 °C and 2 min at 4 °C. Finally, 40 μl water were added; cDNA samples were stored at −20 °C.

To analyse *torso* expression in the isolated fly tissues, cDNAs were PCR-amplified using *torso*-specific primers (see Supplementary Table 1) and a JumpStart REDTaq ReadyMix Reaction Mix (Sigma-Aldrich, MO, USA). α-tubulin was used as internal control. The PCR programme consisted of 5 min at 95 °C, followed by 35 cycles of 30 s at 95 °C, 30 s at 65 °C, 60 s at 72 °C and finally 5 min at 72 °C.

Quantitative PCR with reverse transcription (qRT-PCR) for *torso* expression was carried out using *torso*-specific primers suitable for qRT-PCR (see Supplementary Table 1); *rp49* was used as reference. Reaction mix included 5 μl Maxima SYBR Green/ROX qPCR Master Mix (2 ×)(Thermo Fisher Scientific Inc., Waltham, MA, USA), 0.25 μl primers (10 μM), 1 μl cDNA and 3.5 μl nuclease-free water. PCRs were carried out using an Agilent Mx3000P QPCR System thermocyler (Santa Clara, CA, USA) using the following regime: 25 °C for 5 s, 95 °C for 10 min, followed by 40 cycles of 15 s at 95 °C, 15 s at 57 °C and 15 s at 72 °C. A melting curve was done at the end of the reaction, which consisted of 10 s at 95 °C, 5 s at 25 °C, 5 s at 70 °C and ending with 1 s at 95 °C. All analyses were carried out using MxPro QPCR Software (Agilent, Santa Clara, CA, USA). At least three independently isolated cDNAs were used, and each cDNA was qRT-PCR-amplified in triplicate.

qRT-PCR for *Ptth* expression was carried out using *Ptth*-specific primers suitable for qRT-PCR (see Supplementary Table 1); α-tubulin and *rp49* were used as reference. Reaction mix included 10 μl SensiMix SYBR Green/NoROX qPCR Master Mix (Bioline, Luckenwalde, Germany), 0.8 μl primers (10 μM), 5 μl cDNA and 3.4 μl nuclease-free water. PCRs were carried out using an Eppendorf Mastercycler (Eppendorf, Hamburg, Germany) using the following regime: 95 °C for 2 min, followed by 35 cycles of 5 s at 95 °C, 10 s at 60.5 °C, 15 s at 72 °C. A melting curve was done at the end of the reaction. Primer efficiency was tested by a dilution series and optimal annealing temperature was determined through a gradient PCR. Analyses were carried out using Eppendorf Mastercycler ep realplex software version 2.2. (Eppendorf, Hamburg, Germany) with consideration of the noise band calculation. Five independently isolated cDNAs were used, and each cDNA was qRT-PCR-amplified in triplicate.

**Emergence monitoring.** Flies were raised at 20 °C under 12 h light and 12 h darkness (LD 12:12). After 12–17 days, pupae were collected and fixed on eclosion plates with Elmer's glue and either mounted on Trikinetics eclosion monitors (Trikinetics, MA, USA) or placed in our custom-made WEclMon eclosion monitors (Ruf *et al.*, in prep.); this is an open camera-based system that allows eclosion events to be detected by semi-automated FIJI (ref. 66)-based image analysis. Emergence was then monitored for 7–10 days at 20 °C either under LD 12:12, or DD conditions, in climate and light controlled chambers with ca. 65% relative humidity (Plant Growth Chamber, DR-36NL, Percival Scientific, Inc., Perry, USA and Bioref 19L incubator, PiTec, Chile). Rhythmicity of eclosion profiles was analysed using MATLAB (MathWorks, Inc., Natick, USA) and the appropriate Matlab toolbox[67]. Using autocorrelation analyses, records were considered rhythmic if the rhythmicity index (RI) was ≥ 0.3, weakly rhythmic if 0.1 ≤ RI ≤ 0.3 and arrhythmic if RI < 0.1 (ref. 68).

For experiments involving temperature entrainment, flies were raised under constant red light (635 nm) and a temperature regime of 12 h at 25 °C and 12 h at 16 °C (warm:cool (WC) 12:12). During the shifts, temperature was changed by 0.1 °C per minute. Eclosion was monitored as described above under either WC12:12 or constant temperature (20 °C, CC), and constant red light (635 nm).

**Immunostaining.** Dissected brains or whole flies with holes in thorax and abdomen were fixed in 4% paraformaldehyde for 2 h. Whole flies were then washed with PBS, embedded in 7% agarose, and sectioned using a vibratome (100 μm). Tissues were washed in PBS containing 0.3% TritonX-100 (PBT), blocked with 5% normal goat serum (NGS) in PBT and incubated overnight at 4 °C in primary antibodies. They were then washed in PBT and incubated overnight at 4 °C in secondary antibodies diluted in PBT containing 3% NGS. Finally, specimens were washed, mounted in 80% glycerol in PBS, and stored in darkness until scanning. Projections from PTTH and PDF neurons were visualized in *ptth*-gal4 > 10xmyr-GFP (PTTH) brains using a rabbit anti-GFP antiserum (1:1,000; A6455, Invitrogen, CA, USA) and a mouse anti-PDF monoclonal antibody (1:100; PDF C7 (ref. 69); Developmental Studies Hybridoma Bank, IA, USA, donated by Justin Blau). Secondary antibodies were goat anti-rabbit IgG Alexa Fluor 488 and goat anti-mouse IgG Alexa Fluor 635 (1:250; Invitrogen, MA, USA).

**Image analyses.** A Leica TCS SPE confocal microscope (Leica Microsystems, Wetzlar, Germany) with a × 20 or × 40 high aperture objective was used to scan the labelled specimens with a z-step size from 0.8 to 1.0 μm. Image projections were made using FIJI (ref. 66). Brightness and contrast adjustment were performed with Photoshop CS5 (Adobe Systems, San Jose, CA, USA). Sagittal views (cf., Fig. 1d,e) were recorded under a Leica M165 FC fluorescence stereo microscope (Leica Microsystems, Germany).

**Live imaging of fluorescence.** Pharate flies expressing the calcium sensor GCaMP6m (ref. 70) or the cAMP sensor Epac1-camps[33] in PTTH neurons were dissected in HL3.1 saline[65] and placed at the bottom of the lid of a plastic Petri dish (35 × 10 mm; Greiner Bio-One International GmbH, Austria) filled with HL3.1. Brains were left to adhere to the dish for 15 min then imaged using a widefield fluorescence imaging setup consisting of a motorized Zeiss AxioExaminer D1 upright microscope equipped with a Cairn Twin-Cam LS image splitter, two pco.edge 4.2 sCMOS cameras, and a SPECTRA-4 light engine; or a Zeiss Axioskop FS2 upright microscope equipped with a Photometrics DualView2, a Photometrics CoolSnap CCD camera, and a VisiChrome monochromator (both systems from Visitron, Puchheim, Germany). Images were acquired with a Zeiss W 'Plan-Apochromat' × 20/1.0 or × 40/1.0 water immersion objective and analysed using Visiview 3.0 software (Visitron, Puchheim, Germany).

For GCaMP6 calcium imaging, an excitation wavelength of 475 nm and an exposure time of 100 ms at 4x binning was used. Imaging was performed over at least 600 s with a frame rate of 0.5 Hz using a Chroma ET-GFP emission filter. After 300 s, sNPF-1 (AQRSPSLRLRFamide; Iris Biotech GmbH, Marktredwitz, Germany) dissolved in HL3.1 containing 0.1% DMSO or HL3.1 containing 0.1% DMSO (control) were bath applied. Regions of interest (ROIs) were drawn over the cell bodies of PTTH neurons. Background was subtracted for every ROI and the changes in fluorescence intensity were calculated as:

$$\Delta F/F_0 = (F_n - F_0)/F_0 \qquad (1)$$

where $F_n$ is the fluorescence at time point $n$ and $F_0$ is the baseline fluorescence value calculated from frame 139 to 149 (278–298 s, just before application of the solution). Mean fluorescence intensity change was calculated by subtracting the mean $\Delta F/F_0$ of the first 300 s from the mean $\Delta F/F_0$ of the 300 s after bath application of the solution. Calcium spikes were counted for each graph in the first 300 s (pre) and from 302 to 602 s.

For Epac1-camps imaging, an excitation wavelength of 434/17 nm and an exposure time of 120 ms at 4 × binning (Axioskop) or 200 ms without binning (Examiner) was used. Imaging was performed over at least 1500 s with a frame rate of 0.2 Hz using Chroma ET-CFP and ET-YFP emission filters and a dualband CFP/YFP dichroic mirror. After 325 s PDF (NSELINSLLSLPKNMNDAa; Iris Biotech GmbH, Marktredwitz, Germany) and sNPF, respectively, dissolved in HL3.1 containing 0.1% DMSO or HL3.1 containing 0.1% DMSO (control) was applied. After each experiment, NKH477 (Forskolin analog; Merck Millipore, Darmstadt, Germany), a membrane-permeable activator of adenylyl cyclase, was applied as a positive control. Background subtraction was performed for CFP and YFP intensities of every ROI. The CFP/YFP ratio was calculated for each time point as:

$$\Delta(CFP/YFP) \qquad (2)$$

with

$$CFP = CFP_n/CFP_0 \qquad (3)$$

and

$$YFP = (YFP_n - (CFP_n \times \text{spillover into YFP}))/YFP_0 \qquad (4)$$

$XFP_n$ is the fluorescence at time point $n$ and $XFP_0$ is the baseline fluorescence calculated from frame 2 to 8 (5–35 s). The relative change in cAMP levels after PDF application was calculated as mean of the maximum CFP/YFP change over time in per cent.

**Light stimulation.** To activate *pdf*-LexA-positive neurons expressing LexAop-ChR2-XXL, two 1-min light pulses of 475 nm were applied to pharate brains kept in HL3.1. For GCaMP6 calcium imaging in PTTH neurons, an excitation

wavelength of 475 nm with an intensity of 3780 μW cm$^{-2}$ and an exposure time of 80 ms at 1 × binning was used. Imaging was performed over 1202 s with a frame rate of 0.5 Hz using a Chroma ET-GFP emission filter. After 150 time points (300 s) the imaging was paused and a 1 min light pulse of 7530 μW cm$^{-2}$ was applied to the brain. After 390 time points a second light pulse of 14280 μW cm$^{-2}$ was applied. ROIs were drawn over the cell bodies of PTTH neurons. Mean fluorescence intensity change was calculated by subtracting the mean $\Delta F/F_0$ of the first 300 s from the mean $\Delta F/F_0$ of the 300 s after the second light pulse. Calcium spikes were counted for each graph in the first 300 s (pre) and the 300 s after the second light pulse (after light).

**Imaging of bioluminescence.** Brains expressing *cre*-luc[38] were dissected in ice-cold Schneider's medium (Sigma-Aldrich) containing 1% antibiotic solution (10,000 U ml$^{-1}$ penicillin and 10 mg ml$^{-1}$ streptomycin; Sigma-Aldrich). The brains were carefully placed on poly-lysine coated FluoroDish plates (FD35-100; WPI, FL, USA). They were then covered with Schneider's medium containing 1% antibiotic solution, supplemented with 10% fetal bovine serum and 10 μg ml$^{-1}$ insulin (Sigma-Aldrich), and containing 1 mM luciferin (Potassium Salt, Gold Biotechnology Inc. MO, USA). Preparations were viewed using an LV200 microscope (Olympus, Japan) under 20× magnification and imaged for 72–96 h with an Evolve 512 camera (Photometrics, Tucson, AZ, USA) using 20 min exposures and 200× gain. Records were detrended using FIJI (ref. 66).

**Statistical analyses.** Statistical analyses for Fig. 2 and Supplementary Fig. 2 were performed with R version 3.2.2 (2015-08-14; The R Foundation for Statistical Computing). The imaging data were first analysed for normal distribution using Shapiro-Wilk test. Statistical comparisons between genotypes were then carried out, using *t*-test for normally distributed data and Wilcoxon rank sum test for non-normally distributed data. For analyses of behavioural rhythmicity, values were compared by one-way ANOVA, followed by Tukey's *post hoc* multiple comparison analyses using Prism 6.0 (Graphpad Software Inc, CA). For eclosion studies, three independent replicates are sufficient to define a genotype as rhythmic, weakly rhythmic or arrhythmic. For important genotypes or when an accurate estimate of periodicity is important (for example, for Fig. 8) N was always greater than 3.

**Data availability.** All relevant data are available from the authors upon request.

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

## Acknowledgements

We thank the Bloomington stock centre, the Vienna *Drosophila* resource centre and the following colleagues for flies: Orie Shafer, Naoki Yamanaka and Michael O'Connor, Jerry Yin, Dick Nässel, Todd Holmes, Paul Taghert and Paul Hardin. We thank Michael O'Connor for *phm*- and *ptth*-containing DNA clones, Tobias Langenhan, Robert Kittel and Georg Nagel for a ChR2-XXL-containing DNA clone, and Benjamin White for a gal80-containing DNA clone; Gertrud Gramlich and Susanne Klühspies for excellent technical help. We thank the Developmental Studies Hybridoma Bank for anti-PDF antibodies. We thank Raúl Vallejos for help with construction of *phm*-gal80. We appreciate comments on the manuscript from Charlotte Förster and Paul Taghert, whom

we also thank for suggestions during the course of this work. This research was supported by CONICYT FONDEQUIP grant EQM 120031 (to J.E.), FONDECYT grant 1141278 (to J.E.); Office of Naval Research Global N62909-13-1-N251 and N62909-14-1-N205 (to J.E.) and Centro Interdisciplinario de Neurociencia de Valparaíso (CINV) Millennium Institute grant P09-022-F, supported by the Millennium Scientific Initiative of the Ministerio de Economía, Fomento y Turismo (to J.E.); FONDECYT postdoctoral grant 3090019 (to C.M.); FONDECYT postdoctoral grant 3160177 (to A.P.-M.); the German Research Foundation (Deutsche Forschungsgemeinschaft DFG), collaborative research centre SFB 1047 'Insect timing', project B2 (to C.W.) and the Alexander von Humboldt foundation (to C.I.).

## Author contributions

M.S., C.M., A.P.-M., F.R., L.U., J.C., G.B., C.I., V.S., C.W. and J.E. performed the experiments. M.S., C.M., A.P.-M., F.R., C.W. and J.E. analysed the data. J.E. and C.W. wrote the paper, with feedback from all authors. All authors edited and approved the manuscript.

## Additional information

**Competing interests:** The authors declare no competing financial interests.

