## [Peer Review File · Nature Communications]

Reviewers' comments:

Reviewer #1 (Remarks to the Author):

In this study, the authors revealed that the neural pathway between the brain clock and the PTTH neurons and the PG that controls the timing of eclosion. Since the developmental timing and gating system is widely observed in animals including the birth of baby in humans. Their finding will provide an excellent model to understand how the circadian clock regulates the timing of development, which is a significant contribution in science. Their data are very convincing and statistics are good. They did most of possible experiments that we would expect to do to draw the conclusion. So, as a whole, this work should be appropriate for Nature Communication. However, I have several concerns that the authors should consider in the revised version.

1. The authors concluded that the arrhythmicity of the pdf mutants in eclosion rhythms is due to the disruption of the clock in the mutants. However, there is no evidence for that. They cited one paper from the Paul Taghert group that examined the molecular oscillations in clock neurons in DD in adult brains and revealed that the molecular oscillations of the four s-LNv neurons desynchronize each other in the mutants after prolonged DD conditions, which suggests that the pdf mutants still persist normal molecular oscillations in the beginning of DD as they showed in their paper. Therefore, the results of Lin et al., 2004 even contradict the conclusion that the authors drew. Do the authors have own eclosion data of the pdf01 mutants? Do their eclosions immediately become arrhythmic in DD? If so, there might be an alternative explanation that the PDF signaling affects pathway(s) other than the PTTH neurons, keeping their rhythmicity in DD.

2. Killing and silencing the PTTH neurons resulted in arrhythmicity in statistics but most of them showed rather clear rhythmic peaks in the first days of DD, which means that the rhythms damp in DD. So, I guess that those flies are totally rhythmic in LD. The similar tendency can also be observed in the *ptth>snpr* RNAi flies. Thus, the role of the PTTH neurons could be to mediate between the clock and PG in free-running conditions. On page 7, line 13-15, the authors described "These results demonstrate that the PTTH neurons provide, via the PTTH neuropeptide, the brain signal that gates the time of adult emergence under light entrainment". I do not understand how they can conclude the role of PTTH or the PTTH neurons in light entrainment with DD data. They should explain it logically.

3. On page 7, line 17-18, "However, the role of PTTH in eclosion rhythmicity is not limited to photic information, but extends to temperature entrainment". I do not understand this. The ablation of the PTTH neurons does not affect eclosion rhythms under temperature cycles, right? Thus, I would conclude that pathway other than PTTH is important under temperature cycles. Also, I'm not sure whether these temperature experiments are important in this study. I think that LD data would be more important.

4. To check the clock in the PG, they employed the cre-luc reporter (Fig. 3G). The data are very interesting but the conclusion is very poor. The question is whether the cre-luc rhythms reflect the molecular oscillations in the PG. It could be due to rhythmic input signals from the brain clock. So, I would recommend to do classical immunostaining using a clock protein antibody to look at the clock protein cycling in the PG. Four time points would be sufficient to show that they are rhythmic or not.

5. The authors clearly showed that both of the brain clock and the PG clock are important for eclosion rhythms in DD. However, it is still not clear about the role of the PG clock. It would be great if the authors have any assumption for that. So far, there is not input from the PG to the brain clock. Thus, the brain clock and the PG are one directional relationship but not "master-slave coupling".

Minor comments:

1. Who does belong to Kyoto University?
2. On page 8, line 12, a parenthesis is missing.
3. In Methods, fly rearing and fly stocks. UAS-Kir2.1 and UAS-eko are missing.

4. The legend of Figure 2: are the data from larval brains or pharate adult brains?

Reviewer #2 (Remarks to the Author):

An important question in circadian biology is how central (brain-localized) circadian clocks communicate with the endogenous clocks of peripheral tissues. The study of Selcho et al. makes a significant contribution to this issue by studying neuronal and peripheral clocks in the *Drosophila* model. They show that a specific neuropeptide sNPF, localized to brain PDF-containing clock neurons has a key function in communicating time information to another class of non-clock neurons (PTTH-containing). In turn, PTTH neurons communicate with the prothoracic gland (PG), the steroid-hormone (ecdysone) synthesizing gland in the fruit fly. As decreased ecdysone titers are known to result in adult eclosion, this work provides an elegant framework for understanding the regulation of a circadian event by communicating central and peripheral clocks. It also brings clarity to previous results implicating PDF neurons in the circadian regulation of eclosion.

The authors provide multiple lines of evidence supporting their model:

1. They show that PDF and PTTH projections overlap in the brain suggesting although not proving that they communicate.
2. Application of sNPF, but not PDF, decreases free calcium in PTTH neurons, and optogenetic activation of the PDF neurons mimics this response. This provides compelling evidence that sNPF neurons (PDF cells) can communicate with PTTH neurons.
3. Knockdown of either the sNPF receptor or PTTH in PTTH neurons results in arrhythmic eclosion, providing a link between sNPF and PTTH actions. Killing or silencing the PTTH neurons resulted in the same phenotype, supporting this idea.
4. Knockdown of the PTTH receptor (*torso*) or other elements of *torso* signaling in the PG also results in arrhythmic eclosion and eliminates PG molecular oscillations, providing a link between PTTH and the PG.
5. By manipulating the speeds of brain and PG clocks, using expression of dominant negative doubletime alleles, the authors show that the brain clock has dominates the PG clock in the circadian regulation of eclosion.

The data present in the paper are of high quality, and the figures (including movies) nicely convey the results. Altogether, the authors provide compelling results that describe a neuronal pathway – from brain to PG – that regulates a circadian event. Importantly, they also show how coupled circadian clocks in the brain and PG control this event, an important contribution to a general understanding of how distinct clocks communicate to regulate circadian processes. Indeed, Colin Pittendrigh, one of the founders of contemporary circadian biology, would appreciate these results, given his life-long fascination with understanding coupled oscillators.

Although the paper includes compelling results that support the author's conclusions, I note a few minor issues below that might improve the paper:

1. It isn't obvious how many RNAi lines were employed in certain studies. Are the results from use of single RNAi transgenes or were they replicated with independent RNAi-expressing lines?
2. Was there any quantitation done to show evidence of knockdown for particular RNAi-targeted genes?
3. There are stars or lower case symbols indicating statistical significance in Figures 2 and 3, but I don't see any mention of statistical methods or significance (perhaps I missed it?).

Reviewer #3 (Remarks to the Author):

This is a beautiful study, well conceived, well executed, and written with simple and direct clarity. The authors examined how a circadian clock in the brain regulates a peripheral clock in the prothoracic gland (PG) by a pathway involving the peptide sNPF which stimulates the release of PTTH and in turn the timing of adult emergence via the PG. As noted by the authors there are at least two other examples that indicate a relationship between central and peripheral clocks in the adult fly but this study stands out for several reasons and it would be a mistake simply to lump these studies together. In the big picture, this study takes on a transitional moment in the life of the fly: emergence from the pupal case. The underlying physiology involves a neuroendocrine pathway that regulates the expression of the steroid hormone ecdysone. The involvement of sNPF, as opposed to PDF, is also very interesting. As I read this, I wondered about whether this pathway persists in the control of metabolic events and might even play a role in feeding behaviour later on in the adult. There is an enormous amount of data and each point is well founded and backed up by experiments. The figures are clear. The readers of this journal should be fascinated by this study on the temporal control of the birth of a fly, and readers will be struck by the well documented parallel phenomena in mammals. I urge the editors to accept this study.

Reviewer #4 (Remarks to the Author):

In this excellent manuscript Selcho and colleagues characterize the neurobiological pathway linking the central circadian clock to the peptidergic neurons and steroid producing glands that drive eclosion in the fly *Drosophila melanogaster*. The eclosion rhythm of *Drosophila* was one of the first systematically studied outputs of the circadian clock in animals and its neuronal and endocrine control is understood in remarkable detail, making it a particularly useful rhythm for the study of circadian output. Though the neurobiological control of central timekeeping has made significant strides over the past two decades, the understanding of how time of day information makes its way out of the dedicated timekeeping network through neuronal and endocrine output pathways is not well understood. Is there a single output pathway linking the clock network to systemic time signals that govern the clocks myriad outputs, or are the various outputs of the clock each mediated by their own specific neural and endocrine output pathways? This work represents important progress on several fronts. This study offers a fine characterization of a specific output pathway, revealing the physiological and neurochemical interactions between neurons within the brain to the endocrine and steroid signals linking the brain and peripheral tissues that mediate a specific clock output and shows unequivocally that this pathway is specific to eclosion. The study reveals the hierarchical relationship between the central neuronal oscillators and the peripheral oscillators governing the eclosion rhythm and reveals the requirement for both, a situation remarkably similar to the circadian control of steroid signaling in mammals. Though several recent high profile studies have made some progress on circadian output relevant to locomotor rhythms this study offers the most complete understanding of a circadian output pathway in the fly to date. Furthermore, the question of how central and peripheral circadian oscillators interact to govern rhythms is a centrally important question for which little progress has been made. The discovery that the fly employs a mechanism for the control of steroid signaling that is highly similar to that of mammals is likewise a major contribution. This work is therefore of great interest across several fields of study.

I have only a few relatively minor questions that if addressed would improve this already excellent study.

The study makes excellent use of RNAi knockdown to support their model of the circadian output pathway governing eclosion and have carefully examined the potential non-specific effects of each of the GAL4 drivers and UAS-RNAi elements they have used to create experimental animals. The methods section indicates that a UAS-Dicer2 element was used for each of the RNAi experiments. This element was not mentioned in the results section and does not appear to have been controlled for. How confident are the authors that overexpressing Dicer2 alone in the various cell-types had no effects on eclosion rhythms?

The optogenetic analysis of the connection between the PDF expressing clock neurons and the PTTH expressing cells is an important component of the study and provides strong evidence for a sNPF-mediated inhibitory connection between these cell types. In order to accomplish this, the authors created a new LexAop-ChR-XXL for use with the Pdf-LexA driver. It appears that the LexAop-ChR-XXL element they created were randomly inserted into the genome (i.e., they were not placed in specific, well characterized landing sites). Given the possibility of leaky ChR-XXL expression and the widespread expression of sNPF in the brain, I worry a little that the physiological effects observed in PTTH neurons during the optogenetic experiment designed to excite PDF neurons could be due to leaky expression of the opsin outside of the PDF neurons (randomly inserted coding sequence can sometimes land near enough to native enhancers for leaky expression to be a problem). For this reason, I think that the use of flies containing the LexAop-ChR-XXL element without the Pdf-LexA driver would be a superior control compared to the negative control used in the current manuscript (a fly with the Pdf-LexA driver but without the LexAop-ChR-XXL, if I understand correctly).

I'm also curious about one other aspect of the optogenetics experiment. To measure PTTH neuron responses the authors employ the Ca²⁺ sensor GCaMP, which requires blue light illumination. The authors therefore used 475nm light to visualize GCaMP. In order to optogenetically stimulate PDF neurons, the authors used two one-minute pulses of 475nm light during GCaMP imaging. Why didn't GCaMP imaging by itself cause the opening of ChR-XXL, the excitation of PDF neurons, and the inhibition of PTTH neurons? Is this simply a matter of different light intensities being used? The authors should supply an explanation for how they arrived at this particular approach and how they were able to ensure that opsins were not activated by imaging alone. As written it is difficult to imagine how one would replicate this critical experiment. Also, the authors state that they stopped GCaMP imaging during the optogenetic light pulses, presumably to avoid a large artefactual increase in GCaMP fluorescence. However, these breaks in imaging are not present in the GCaMP plots. It is not clear why they should not be seen. Were time-points during the stimulation simply omitted? Is the X-axis not an unbroken series of time-points?

Quibbles:

On the bottom of page 9 the authors refer to the specific slowing down of the brain clock. But in fact they are using the Tim-GAL4 driver with the phm-GAL80 element. This would be predicted to slow down all clocks except those of the PG, no? There are drivers that could be employed to specifically affect neurons if this was the goal. I suggest re-wording things here for accuracy.

I'm not aware of a widely accepted "standard" fly media. The authors should describe what their fly food was made of.

There are reportedly up to 4 different possible forms of sNPF, what was the sequence of the sNPF peptide used in this study?

A brief description of the WEClMon system would be helpful.

Response to Reviewers

Reviewer #1 (Remarks to the Author):

In this study, the authors revealed that the neural pathway between the brain clock and the PTTH neurons and the PG that controls the timing of eclosion. Since the developmental timing and gating system is widely observed in animals including the birth of baby in humans. Their finding will provide an excellent model to understand how the circadian clock regulates the timing of development, which is a significant contribution in science. Their data are very convincing and statistics are good. They did most of possible experiments that we would expect to do to draw the conclusion. So, as a whole, this work should be appropriate for Nature Communication. However, I have several concerns that the authors should consider in the revised version.

We appreciate this reviewer's overall positive assessment of our manuscript.

1. The authors concluded that the arrhythmicity of the *pdf* mutants in eclosion rhythms is due to the disruption of the clock in the mutants. However, there is no evidence for that. They cited one paper from the Paul Taghert group that examined the molecular oscillations in clock neurons in DD in adult brains and revealed that the molecular oscillations of the four s-LNv neurons desynchronize each other in the mutants after prolonged DD conditions, which suggests that the *pdf* mutants still persist normal molecular oscillations in the beginning of DD as they showed in their paper. Therefore, the results of Lin et al., 2004 even contradict the conclusion that the authors drew. Do the authors have own eclosion data of the *pdf01* mutants? Do their eclosions immediately become arrhythmic in DD? If so, there might be an alternative explanation that the PDF signaling affects pathway(s) other than the PTTH neurons, keeping their rhythmicity in DD.

We agree that our description of the phenotype of *pdf* mutants is an oversimplification because this genotype does indeed express a rhythm during the first 1-2 days in DD, which then degrades into arrhythmicity. We have modified the relevant section of text, and taken this opportunity to more accurately describe the emergence phenotype of this genotype with additional text (p. 6) and a Supplementary figure (Supplementary Fig. 1).

2. Killing and silencing the PTTH neurons resulted in arrhythmicity in statistics but most of them showed rather clear rhythmic peaks in the first days of DD, which means that the rhythms damp in DD. So, I guess that those flies are totally rhythmic in LD. The similar tendency can also be observed in the *ptth>snpfr* RNAi flies. Thus, the role of the PTTH neurons could be to mediate between the clock and PG in free-running conditions.

The presence of peaks of emergence during the first 1-2 days in DD for genotypes in which we manipulated PTTH neurons (either silencing or killing them, or expressing sNPFR RNAi) is a very acute observation. By superimposing the eclosion profiles from several experiments we confirmed that these genotypes do indeed all show these initial peaks. Nevertheless, we have not included these data because we are not 100% confident that these records accurately reflect the phenotype of these genotypes because the PTTH-gal4 driver we used (best one available) is not very strong. Thus there is a possibility that the phenotypes we observe are those of hypomorphs, and might differ from those of animals completely lacking the corresponding function (e.g., a null PTTH mutant). Nevertheless, we have included this analysis for *pdf* mutants (see response to comment #1, above) and for animals expressing *torso* RNAi in the PG (Supplementary Fig. 1), as we are quite confident about these genotypes: indeed, the

pdf mutants are null alleles, and expressing *torso* RNAi in the PG consistently produced a completely arrhythmic phenotype as well as a dramatic decrease in *torso* RNA expression (Supplementary Fig. 6C,D).

On page 7, line 13-15, the authors described “These results demonstrate that the PTTH neurons provide, via the PTTH neuropeptide, the brain signal that gates the time of adult emergence under light entrainment”. I do not understand how they can conclude the role of PTTH or the PTTH neurons in light entrainment with DD data. They should explain it logically.

We agree that this was not the correct phrasing and changed the sentence (p. 7; see also point 3, below).

3. On page 7, line 17-18, “However, the role of PTTH in eclosion rhythmicity is not limited to photic information, but extends to temperature entrainment”. I do not understand this. The ablation of the PTTH neurons does not affect eclosion rhythms under temperature cycles, right? Thus, I would conclude that pathway other than PTTH is important under temperature cycles. Also, I’m not sure whether these temperature experiments are important in this study. I think that LD data would be more important.

We agree that this was not the correct phrasing. We found that eclosion was arrhythmic under constant conditions after temperature entrainment, as it was after LD entrainment. This suggests that PTTH is required for rhythmicity rather than for simply transmitting to the PG a rhythmic signal. On the other hand eclosion was rhythmic when monitored under LD or WC conditions, which is likely due to the entrainment of the PG by light and temperature. We have changed the text (p. 7) and also included the data obtained under a LD regime (Fig. 4C), following the reviewer’s suggestion.

4. To check the clock in the PG, they employed the *cre-luc* reporter (Fig. 3G). The data are very interesting but the conclusion is very poor. The question is whether the *cre-luc* rhythms reflect the molecular oscillations in the PG. It could be due to rhythmic input signals from the brain clock. So, I would recommend to do classical immunostaining using a clock protein antibody to look at the clock protein cycling in the PG. Four time points would be sufficient to show that they are rhythmic or not.

We understand this reviewer’s concern. Although several clock gene-luciferase transgenic lines exist (Stanewsky, R. (2007). Analysis of rhythmic gene expression in adult *Drosophila* using the firefly luciferase reporter gene. *Methods Mol Biol* **362**, 131-142) in our hands none produced robust bioluminescence from the PG. And despite repeated attempts with various available anti-PER and anti-TIM antibodies, we were not able to obtain consistent immunostaining in the PG. For this reason we settled on using *cre-luc* expression, because it has been used as a proxy for clock activity (Belvin, M.P., Zhou, H., and Yin, J.C. (1999). The *Drosophila* dCREB2 gene affects the circadian clock. *Neuron* **22**, 777-787) and produced the most reliable bioluminescence from the PG. Nevertheless, we did attempt to address the concern raised by this reviewer, and now include traces from a *period-luc* line in animals expressing *torso* RNAi in the PG and in corresponding controls (Supplementary Fig. 7). Although cycling is less robust than using the *cre-luc* line a similar result was obtained, with *torso* knockdown in the PG eliminating cycling (as well as causing an increased amplitude signal).

5. The authors clearly showed that both of the brain clock and the PG clock are important for eclosion rhythms in DD. However, it is still not clear about the role of the PG clock. It would be great if the authors have any assumption for that. So far, there is not input from the PG to the

brain clock. Thus, the brain clock and the PG are one directional relationship but not “master-slave coupling”.

We would indeed love to better understand the role of the PG clock! All we can conclude from our findings is that both central and PG clocks are necessary for circadian rhythmicity of emergence. We speculate in the Discussion (pp. 12-13) that the contribution of the PG clock may be in regulating the timing of the final fall in ecdysteroid (E) titers. Nevertheless, accurate measurements of these titers during the last 2 days of metamorphosis have not been done for *Drosophila*, so this proposal remains in the realm of speculations. Progress forward will likely come from investigating the exact role of genes involved in steroidogenesis or E action that are directly regulated by the clock, including *Eip75B* (the fly homolog of REV-ERB α and REV-ERB β , which are key regulatory elements of the core molecular clock of mammals).

Minor comments:

1. Who does belong to Kyoto University? Thank you for pointing out this error. Chihiro Ito now has a “Current address”.
2. On page 8, line 12, a parenthesis is missing. Thank you! This has been fixed.
3. In Methods, fly rearing and fly stocks. UAS-Kir2.1 and UAS-eko are missing. Thank you! They have been added to the list of stains used.
4. The legend of Figure 2: are the data from larval brains or pharate adult brains? They were pharate adult brains. This has now been indicated in the legend to Figure 2.

Reviewer #2 (Remarks to the Author):

An important question in circadian biology is how central (brain-localized) circadian clocks communicate with the endogenous clocks of peripheral tissues. The study of Selcho et al. makes a significant contribution to this issue by studying neuronal and peripheral clocks in the *Drosophila* model. They show that a specific neuropeptide sNPF, localized to brain PDF-containing clock neurons has a key function in communicating time information to another class of non-clock neurons (PTTH-containing). In turn, PTTH neurons communicate with the prothoracic gland (PG), the steroid-hormone (ecdysone) synthesizing gland in the fruit fly. As decreased ecdysone titers are known to result in adult eclosion, this work provides an elegant framework for understanding the regulation of a circadian event by communicating central and peripheral clocks. It also brings clarity to previous results implicating PDF neurons in the circadian regulation of eclosion.

The authors provide multiple lines of evidence supporting their model:

1. They show that PDF and PTTH projections overlap in the brain suggesting although not proving that they communicate.
2. Application of sNPF, but not PDF, decreases free calcium in PTTH neurons, and optogenetic activation of the PDF neurons mimics this response. This provides compelling evidence that sNPF neurons (PDF cells) can communicate with PTTH neurons.
3. Knockdown of either the sNPF receptor or PTTH in PTTH neurons results in arrhythmic eclosion, providing a link between sNPF and PTTH actions. Killing or silencing the PTTH neurons resulted in the same phenotype, supporting this idea.
4. Knockdown of the PTTH receptor (torso) or other elements of torso signaling in the PG also results in arrhythmic eclosion and eliminates PG molecular oscillations, providing a link between PTTH and the PG.
5. By manipulating the speeds of brain and PG clocks, using expression of dominant negative doubletime alleles, the authors show that the brain clock has dominates the PG clock in the circadian regulation of eclosion.

The data present in the paper are of high quality, and the figures (including movies) nicely convey the results. Altogether, the authors provide compelling results that describe a neuronal pathway – from brain to PG – that regulates a circadian event. Importantly, they also show how coupled circadian clocks in the brain and PG control this event, an important contribution to a general understanding of how distinct clocks communicate to regulate circadian processes. Indeed, Colin Pittendrigh, one of the founders of contemporary circadian biology, would appreciate these results, given his life-long fascination with understanding coupled oscillators.

Although the paper includes compelling results that support the author's conclusions, I note a few minor issues below that might improve the paper:

We appreciate this reviewer's overall positive view of our work, and were especially happy to read that they recognized the historical significance of insect emergence rhythms to circadian biology.

1. It isn't obvious how many RNAi lines were employed in certain studies. Are the results from use of single RNAi transgenes or were they replicated with independent RNAi-expressing lines?

In preliminary experiments we tested several UAS-RNAi lines (2-5). In most cases a range of phenotypes was obtained and we chose the line that produced the most extreme one. A line to this effect has been added to the Methods section (p. 15)

2. Was there any quantitation done to show evidence of knockdown for particular RNAi-targeted genes?

This measurement was done for *torso* (cf. Supplementary Fig. 6C) where a greater than 10-fold knockdown was achieved. However, for most genetic manipulations this would be a very difficult measurement to conduct since the target gene was expressed quite broadly. For instance, many brain neurons express sNPF suggesting that its receptor (sNPFR) will also be broadly expressed. Thus, selective knockdown on sNPFR only in the (4) PTTH neurons would likely not be detectable by most relatively simple quantitative methods such as qRT-PCR. Nevertheless, for cases where no effect was observed we used other means to show that this was indeed because of the lack of expression, and not due to lack of effective RNAi knockdown. For instance, we show that PTTH neurons do not respond to the PDF neuropeptide unless the PDFR is expressed in these neurons (Fig. 2A), suggesting that the lack of defects caused by PDFR knockdown (Fig. 3B) is due to the lack of PDFR in PTTH neurons, not to our inability to cause significant knockdown of PDFR function. In addition, the fact that several different RNAi lines produced qualitatively similar results (see point 1, above) suggest that the phenotypes obtained were caused by the knockdown of the target gene, not to RNA interference of non-target genes.

3. There are stars or lower case symbols indicating statistical significance in Figures 2 and 3, but I don't see any mention of statistical methods or significance (perhaps I missed it?).

We thank the reviewer for pointing out this omission. We have added the relevant details to the legend of Figure 2; we have also added details on the statistical analyses performed in the Methods section (pp. 24-25).

Reviewer #3 (Remarks to the Author):

This is a beautiful study, well conceived, well executed, and written with simple and direct clarity. The authors examined how a circadian clock in the brain regulates a peripheral clock in the prothoracic gland (PG) by a pathway involving the peptide sNPF which stimulates the release of PTTH and in turn the timing of adult emergence via the PG. As noted by the authors there are at least two other examples that indicate a relationship between central and peripheral clocks in the adult fly but this study stands out for several reasons and it would be a mistake simply to lump these studies together. In the big picture, this study takes on a transitional moment in the life of the fly: emergence from the pupal case. The underlying physiology involves a neuroendocrine pathway that regulates the expression of the steroid hormone ecdysone. The involvement of sNPF, as opposed to PDF, is also very interesting. As I read this, I wondered about whether this pathway persists in the control of metabolic events and might even play a role in feeding behaviour later on in the adult. There is an enormous amount of data and each point is well founded and backed up by experiments. The figures are clear. The readers of this journal should be fascinated by this study on the temporal control of the birth of a fly, and readers will be struck by the well documented parallel phenomena in mammals. I urge the editors to accept this study.

We appreciate the enthusiastic review!

Reviewer #4 (Remarks to the Author):

In this excellent manuscript Selcho and colleagues characterize the neurobiological pathway linking the central circadian clock to the peptidergic neurons and steroid producing glands that drive eclosion in the fly *Drosophila melanogaster*. The eclosion rhythm of *Drosophila* was one of the first systematically studied outputs of the circadian clock in animals and its neuronal and endocrine control is understood in remarkable detail, making it a particularly useful rhythm for the study of circadian output. Though the neurobiological control of central timekeeping has made significant strides over the past two decades, the understanding of how time of day information makes its way out of the dedicated timekeeping network through neuronal and endocrine output pathways is not well understood. Is there a single output pathway linking the clock network to systemic time signals that govern the clocks myriad outputs, or are the various outputs of the clock each mediated by their own specific neural and endocrine output pathways? This work represents important progress on several fronts. This study offers a fine characterization of a specific output pathway, revealing the physiological and neurochemical interactions between neurons within the brain to the endocrine and steroid signals linking the brain and peripheral tissues that mediate a specific clock output and shows unequivocally that this pathway is specific to eclosion. The study reveals the hierarchical relationship between the central neuronal oscillators and the peripheral oscillators governing the eclosion rhythm and reveals the requirement for both, a situation remarkably similar to the circadian control of steroid signaling in mammals. Though several recent high profile studies have made some progress on circadian output relevant to locomotor rhythms this study offers the most complete understanding of a circadian output pathway in the fly to date. Furthermore, the question of how central and peripheral circadian oscillators interact to govern rhythms is a centrally important question for which little progress has been made. The discovery that the fly employs a mechanism for the control of steroid signaling that is highly similar to that of mammals is likewise a major contribution. This work is therefore of great interest across several fields of study.

We appreciate this reviewer's positive evaluation of our work.

I have only a few relatively minor questions that if addressed would improve this already excellent study.

The study makes excellent use of RNAi knockdown to support their model of the circadian output pathway governing eclosion and have carefully examined the potential non-specific effects of each of the GAL4 drivers and UAS-RNAi elements they have used to create experimental animals. The methods section indicates that a UAS-Dicer2 element was used for each of the RNAi experiments. This element was not mentioned in the results section and does not appear to have been controlled for. How confident are the authors that overexpressing Dicer2 alone in the various cell-types had no effects on eclosion rhythms?

We appreciate the reviewer's concern but are quite confident that the defects were observed are not caused by the expression of *dcr2* because the controls for the GAL4 drivers shown in Figure 3 include *dcr2* (i.e., *ptth>pdfr* RNAi was shorthand for *ptth-GAL4+UAS-dcr2+UAS-pdfr*). This detail has been added to the legend of Figure 3. In addition we have indicated in other relevant figure legends (to Fig. 6; and supplementary figures 1, 6, 7 and 8) that all experiments involving RNAi knockdown included the use of *dcr2*.

The optogenetic analysis of the connection between the PDF expressing clock neurons and the PTTH expressing cells is an important component of the study and provides strong evidence for a sNPF-mediated inhibitory connection between these cell types. In order to accomplish this, the authors created a new LexAop-ChR-XXL for use with the Pdf-LexA driver. It appears that the LexAop-ChR-XXL element they created were randomly inserted into the genome (i.e., they were not placed in specific, well characterized landing sites). Given the possibility of leaky ChR-XXL expression and the wide-spread expression of sNPF in the brain, I worry a little that the physiological effects observed in PTTH neurons during the optogenetic experiment designed to excited PDF neurons could be due to leaky expression of the opsin outside of the PDF neurons (randomly inserted coding sequence can sometimes land near enough to native enhancers for leaky expression to be a problem). For this reason, I think that the use of flies containing the LexAop-ChR-XXL element without the Pdf-LexA driver would be a superior control compared to the negative control used in the current manuscript (a fly with the Pdf-LexA driver but without the LexAop-ChR-XXL, if I understand correctly).

We are very grateful for this comment as it revealed that we made a mistake in our description of the genotypes used; we apologize for this. We fully agree that it is necessary to control for LexAop-ChR-XXL leakiness.

The movie was correctly named (*ptth>GCaMP6m, >LexAop-ChR-XXL*), but the legend was incorrect. We are very sorry about this mistake and have changed the main text (top of p: 7), the text to the legend to Figure 2 (p. 34) and to the legend of Supplementary Movie 3 (reproduced below, since Legends to movies are included in the Cover letter to the Editor, following *Nature Communications* instructions)

Supplementary Movie 3. No changes in GCaMP fluorescence can be induced by light pulses of 475nm in the absence of *pdf-LexA* driver. Imaging calcium levels in PTTH neurons expressing GCaMP6 while applying light pulses to a LexAop-ChR2-XXL-bearing pharate brain lacking *pdf-LexA* driver. Light pulses of an intensity of 7530 $\mu\text{W}/\text{cm}^2$ (first light pulse after 300s) and 14280 $\mu\text{W}/\text{cm}^2$ (second light pulse after 780s) at 475nm, respectively (as used for Supplementary Movie 2), did not elicit a significant reduction in GCaMP6 fluorescence in PTTH neurons. Total duration of imaging = 1202s.

I'm also curious about one other aspect of the optogenetics experiment. To measure PTTH neuron responses the authors employ the Ca²⁺ sensor GCaMP, which requires blue light illumination. The authors therefore used 475nm light to visualize GCaMP. In order to optogenetically stimulate PDF neurons, the authors used two one-minute pulses of 475nm light during GCaMP imaging. Why didn't GCaMP imaging by itself cause the opening of ChR-XXL, the excitation of PDF neurons, and the inhibition of PTTH neurons? Is this simply a matter of different light intensities being used? The authors should supply an explanation for how they arrived at this particular approach and how they were able to ensure that opsins were not activated by imaging alone. As written it is difficult to imagine how one would replicate this critical experiment. Also, the authors state that they stopped GCaMP imaging during the optogenetic light pulses, presumably to avoid a large artefactual increase in GCaMP fluorescence. However, these breaks in imaging are not present in the GCaMP plots. It is not clear why they should not be seen. Were time-points during the stimulation simply omitted? Is the X-axis not an unbroken series of time-points?

The reviewer brings up an important question, and when we started the experiments we were also a bit puzzled that it worked as described. We did not discuss this issue in depth in the manuscript to avoid confusion, but we did measure carefully the light intensities at our setup and included this information in the Methods section (p. 23)

Imaging: excitation wavelength of 475 nm with an intensity of 3780 $\mu\text{W}/\text{cm}^2$ (= 37.8 $\mu\text{W}/\text{mm}^2$) and an exposure time of 80 ms.

First light pulse for optogenetic activation: 7530 $\mu\text{W}/\text{cm}^2$ (75.30 $\mu\text{W}/\text{mm}^2$) for 60,000 ms.

Second light pulse for optogenetic activation: 14280 $\mu\text{W}/\text{cm}^2$ (142.8 $\mu\text{W}/\text{mm}^2$) for 60,000 ms.

The light intensity during imaging could not be reduced further as the *ptth*-GAL4 driver is rather weak and a lower intensity would have caused the GCaMP signal to fall below imaging threshold. When we compare 37.8 $\mu\text{W}/\text{mm}^2$ (= 3.7×10^{-2} mW/mm^2) to the data from the original characterization of ChR-XXL light sensitivity in various behavioral assays (Dawydow, A., Gueta, R., Ljaschenko, D., Ullrich, S., Hermann, M., Ehmann, N., Gao, S., Fiala, A., Langenhan, T., Nagel, G., and Kittel, R.J. (2014). Channelrhodopsin-2-XXL, a powerful optogenetic tool for low-light applications. *Proc Natl Acad Sci U S A* **111**, 13972-13977; Fig. 2 and Fig. 4; see Fig 2B as an example below), we expected this light intensity to be enough to fully activate ChR-XXL without retinal. Indeed, we found a steady decrease of Ca²⁺-signals in active PTTH neurons during imaging without giving light pulses of higher intensities. However, the decrease in Ca²⁺ accumulated very slowly, whereas with the higher intensities given over considerably longer time we observed a sudden decline in Ca²⁺. The fact that we only saw a very slowly accumulating Ca²⁺ response (decrease in Ca²⁺ in PTTH neurons after optogenetic stimulation of PDF neurons) at the light conditions used for imaging (which in Dawydow et al. caused maximum response) is –we think- due to the short illumination time and low imaging frequency (80 ms/0.5 Hz) and the requirement of large and >micro-domain Ca²⁺ increases for peptide release (i.e. sNPF from sLNvs), which then indirectly caused the changes in Ca²⁺ signals measured in the downstream PTTH neurons. In any case, we were able to suddenly and strongly decrease Ca²⁺ after strong and long-duration (1 min) activation of ChR-XXL in short-term experiments taking 10-20 min. This response is very robust.

Fig. 2B from Dawydow et al. 2014

Quibbles:

On the bottom of page 9 the authors refer to the specific slowing down of the brain clock. But in fact they are using the Tim-GAL4 driver with the phm-GAL80 element. This would be predicted to slow down all clocks except those of the PG, no? There are drivers that could be employed to specifically affect neurons if this was the goal. I suggest re-wording things here for accuracy.

We agree completely with this comment and this fact had been included but only in the legend to the relevant figure (previously Supplementary Fig. 7, now Supplementary Fig. 9). We have reworded the main text for accuracy (p. 9).

I'm not aware of a widely accepted "standard" fly media. The authors should describe what their fly food was made of.

We agree that our flies would not call the gruel we feed them "standard media". We have added some details of our media's ingredients in the Methods section (p. 14).

There are reportedly up to 4 different possible forms of sNPF, what was the sequence of the sNPF peptide used in this study?

The sequence of the exact peptide used (AQRSPSLRLRF-NH₂) is indicated in the Methods section under "Live imaging of fluorescence" (p. 22).

A brief description of the WEclMon system would be helpful.

A brief description of the WEclMon system has been added to the Methods section (p. 20).

REVIEWERS' COMMENTS:

Reviewer #1 (Remarks to the Author):

The authors perfectly addressed to our comments !! This paper must be ready to be published.

Reviewer #4 (Remarks to the Author):

The authors have directly addressed all of my concerns. This is an outstanding study touching on several central issues in the field. This is outstanding work that deserves broad exposure.